



# TheDiaTo (v1.0) – A new diagnostic tool for water, energy and entropy budgets in climate models

Valerio Lembo[1], Frank Lunkeit[1], and Valerio Lucarini[1,2,3]

[1]Meteorologisches Institut, Universität Hamburg, Hamburg, Germany
[2]Department of Mathematics and Statistics, University of Reading, Reading, UK
[3]Centre for the Mathematics of Planet Earth, Department of Mathematics and Statistics, University of Reading, Reading, UK

*Correspondence to:* Valerio Lembo (valerio.lembo@uni-hamburg.de)

**Abstract.** This work presents Thermodynamic Diagnostic Tool (TheDiaTo), a novel diagnostic tool for studying the thermodynamics of the climate systems with a wide range of applications, from sensitivity studies to model tuning. It includes a number of modules for assessing the internal energy budget, the hydrological cycle, the Lorenz Energy Cycle and the material entropy production, respectively. The routine receives as inputs energy fluxes at surface and at the Top-of-Atmosphere (TOA), for the computation of energy budgets at Top-of-Atmosphere (TOA), at the surface, and in the atmosphere as a residual.

Meridional enthalpy transports are also computed from the divergence of the zonal mean energy budget fluxes; location and intensity of peaks in the two hemispheres are then provided as outputs. Rainfall, snowfall and latent heat fluxes are received as inputs for computing the water mass and latent energy budgets. If a land-sea mask is provided, the required quantities are separately computed over continents and oceans. The diagnostic tool also computes the Lorenz Energy Cycle (LEC) and its

storage/conversion terms as annual mean global and hemispheric values. In order to achieve this, one needs to provide as input three-dimensional daily fields of horizontal wind velocity and temperature in the troposphere. Two methods have been implemented for the computation of the material entropy production, one relying on the convergence of radiative heat fluxes in the atmosphere (indirect method), one combining the irreversible processes occurring in the climate system, particularly heat fluxes in the boundary layer, the hydrological cycle and the kinetic energy dissipation as retrieved from the residuals of the

LEC. A version of these diagnostics has been developed as part of the Earth System Model eValuation Tool (ESMValTool) v2.0a1, in order to assess the performances of CMIP6 model simulations, and will be available in the next release of the tool. The aim of this software is to provide a comprehensive picture of the thermodynamics of the climate system as reproduced in the state-of-the-art coupled general circulation models. This can prove useful for better understanding anthropogenic and natural climate change, paleoclimatic climate variability, and climatic tipping points.

# 1   Introduction

The climate can be seen as a forced and dissipative non-equilibrium system exchanging energy with the external environment, and setting a complex mixture of fluid flows into motion by converting available potential into mechanical energy through a vast range of nonlinear processes. The source of available potential energy is the inhomogeneous absorption of solar radiation. The kinetic energy is eventually dissipated through viscous stress and converted back into heat. Such processes can be described



by taking advantage of the theory of non-equilibrium thermodynamics of continuous multiphase media, and, in particular, of fluids. Non-equilibrium systems are characterised by the presence of, possibly fluctuating, fluxes of matter, chemical species, and energy. The steady state is reached as a result of a possibly complex balance of positive and negative feedbacks, and through the interplay of processes having very diverse time scales and physical underpinning mechanisms. Indeed, the climate

can be viewed in such framework, as its observed variability extends over many orders of magnitude in terms of both spatial and temporal scales, and is divided in already extremely complex subdomains - the atmosphere, the ocean, the criosphere, the biosphere, the active soil - which have themselves very diverse characteristic internal time-scales and are nonlinearly coupled (Peixoto and Oort, 1992; Lucarini et al., 2014).

It is a major endeavour of contemporary science to improve our understanding of the climate system both in the context of

the past, present, and projected future conditions. This is key for understanding, as far as the past goes, the co-evolution of life and of the physico-chemical properties of the ocean, soil, and atmosphere, and for addressing the major challenge faced by our planet as a result of the current anthropogenic climate change. Radically advancing the skills of climate models is key to nearing these goals, and, indeed, efforts aimed in this direction have been widely documented (see the related chapter on the last report, of the Intergovernmental Panel on Climate Change (2013)). Intercomparing and validating climate models

is far from being a trivial task, also at a purely conceptual level (Lucarini, 2013; Frigg et al., 2015). Difficulties can only increase when looking at the practical side of things: how to choose meaningful metrics to study the performance of climate models? Should they be motivated in terms of basic processes of the climate system, or in terms of relevance for an end-user of climate services? In order to put some order in this conundrum, the community of climate modellers has developed a set of standardized metrics for intercomparing and validating climate models (Eyring et al., 2016a). For obvious reasons, the choice

made for such metrics has been biased - for possibly good reasons - in the direction of providing useful information for users. What we propose in this paper is a novel software, prepared according to the required standards mentioned above, capable of providing an integrated perspective on the problem of models validation and intercomparison. The goal here is to look at models through the lens of their dynamics and thermodynamics, in the view of enunciated above ideas about complex non-equilibrium systems. The metrics that we here propose are based on the analysis of the energy and water budgets and transports,

of the energy transformations, and of the entropy production. We summarize below some of the key concepts behind our work.

## 1.1 Energy

A requirement for a non-equilibrium system in contact with an external environment to be at steady state is the presence of vanishing - on the average - energy budget. Inconsistencies in the overall energy budget of long-term stationary simulations have been carefully evaluated (Lucarini and Ragone, 2011; Mauritsen et al., 2012) and various aspects of the radiative and

heat transfers within the atmosphere and between the atmosphere and the oceans have been reconsidered in order to constrain models to a realistic climate (Wild et al., 2013; Loeb et al., 2015). A substantial bias in the energy budget of the atmosphere has been identified in many GCMs, resulting from either the imperfect closure of the kinetic energy budget (Lucarini and Ragone, 2011) or the imperfect closure of the mass balance in the hydrological cycle (Liepert and Previdi, 2012). This picture is made even more complicated by the difficult task of having an accurate observational benchmark of the Earth's energy budget (e.g.





Loeb et al. (2009); von Schuckmann et al. (2016)). In general, an increasing consensus is being reached on the fact that the improvement of climate models requires improving the energetic consistency of the modeled system (Hansen et al., 2011; Lucarini et al., 2011, 2014). Rather than a proxy of a changing climate, surface temperatures and precipitation changes should be better viewed as a consequence of a non-equilibrium steady state system which is responding to a radiative energy imbalance

through a complex interaction of feedbacks. A changing climate, under the effect of an external transient forcing, can only be properly addressed if the energy imbalance, and the way it is transported within the system and converted into different forms is taken into account. The models' skill to represent the history of energy and heat exchanges in the climate system has been assessed by comparing numerical simulations against available observations, where available (Allan et al., 2014; Smith et al., 2015), including the fundamental problem of ocean heat uptake (Exarchou et al., 2015).

A key element in defining the steady-state of the climate system is the balance between the convergence of the horizontal (mostly meridional) enthalpy fluxes due to the atmosphere and the ocean and the radiative imbalance at the top of the atmosphere (TOA). Such imbalance is positive in the low latitudes and negative in the high latitudes, and a compensating transport must be present in order to ensure steady state. As a result, the transport reduces dramatically the meridional temperature gradient with respect to what would be set by radiative-convective equilibrium (Manabe and Wetherald, 1967), i.e. in absence of

large scale atmospheric and oceanic transport. Differences in the boundary conditions, in the forcing and dissipative processes, and in chemical and physical properties of the atmosphere and ocean lead to a specific partitioning of the enthalpy transport between the two geophysical fluids; a different partitioning associated to the same total transport would lead to rather different climate conditions (Rose and Ferreira, 2013; Knietzsch et al., 2015). The role of meridional heat transports in different paleoclimate scenarios and in relation to different forcing has been addressed in various studies (see e.g. Caballero and Langen

(2005); Fischer and Jungclaus (2010)). State-of-the-art climate models have been critically analyzed - with mixed findings across the available models - in order to understand whether they are able to represent correctly the global picture as well as the details of the atmospheric and oceanic heat transports (Lucarini et al., 2011, 2014; Lembo et al., 2016).

In order to understand how the heat is transported by the geophysical fluids, one should clarify what sets them into motion. We focus here on the atmosphere. A comprehensive view of the energetics fuelling the general circulation is given by the

Lorenz Energy Cycle (LEC) framework (Lorenz, 1955; Ulbrich and Speth, 1991). This provides a picture of the various processes responsible for conversion of available potential energy (APE), i.e. the excess of potential energy with respect to a state of thermodynamic equilibrium (cfr. Tailleux (2013) for a review), into kinetic energy and dissipative heating. Under stationary conditions, the dissipative heating exactly equals the mechanical work performed by the atmosphere. In other words, the LEC formulation allows to constrain the atmosphere to the first law of thermodynamics, and the system as a whole can be

seen as a pure thermodynamic heat engine under dissipative non-equilibrium conditions (Ambaum, 2010). The strength of the LEC, or in other words the intensity of the conversion of available potential energy (APE) into kinetic energy (KE), has been evaluated in observational-based datasets (Li et al., 2007; Pan et al., 2017) and climate models (Lucarini et al., 2010a; Marques et al., 2011), mostly ranging from 1.5 to 3.5 $W/m^2$. A considerable source of uncertainty is the hydrostatic assumption, on which the LEC formulation relies, possibly leading to significant underestimation of the kinetic energy dissipation (Pauluis

and Dias, 2012). In some respects, one can see the usual formulation of the LEC as suitable for describing energy exchanges



and transformation at a coarse-grained level, where non-hydrostatic processes are not relevant (and the the system is accurately described by primitive equations). Given its nature of heat engine, an efficiency can also be attributed to the atmosphere, assuming that the system is analogous to an engine working operating a warm and a cold temperature (Lucarini, 2009). This approach has been generalized by Pauluis (2011), in order to account for the role of water vapor.

## 1.2 Water

Indeed, water is an essential ingredient of the climate system, and the hydrological cycle has an important role on the energy pathways of the climate system, because water vapour and clouds impact the radiative processes inside the system, and water phase exchanges are extremely energy intensive. Similarly to the case of energy imbalances, a closed water-mass conserving reproduction of the hydrological cycle has proved to be essential, not only because of the disparate implications of the hydrological cycle for energy balance and transports in the atmosphere, but also because of its sensitivity to climate change (Held and Soden, 2006) and the extreme relevance of the cloud and water vapor feedbacks (Hartmann, 1994). On one hand the energy budget is relevantly affected by semi-empirical formulations of the water vapor spectrum (Wild et al., 2006), on the other hand the energy budget influences the moisture budget by means of uncertainties in aerosol-cloud interactions and mechanisms of tropical deep convection (Wild and Liepert, 2010; Liepert and Previdi, 2012). A global scale evaluation of the hydrological cycle, both from a moisture and energetic perspective, is thus considered an integral part of an overall diagnostics for the thermodynamics of climate system.

## 1.3 Entropy

The climate system has long been recognized as featuring irreversible processes through dissipation and mixing in various forms, leading to the production of entropy (Paltridge, 1975), which is one of the key properties of non-equilibrium systems (Prigogine, 1962). Several early attempts (Peixoto and Oort, 1992; Johnson, 1997; Goody, 2000) have been made to understand the complex nature of irreversible climatic processes. Recent works (Bannon, 2015; Bannon and Lee, 2017) have proposed an innovative approach by partitioning the system into a control volume made of matter and radiation, which exchanges energy with its surroundings. Raymond (2013) has described the entropy budget of an aggregated dry air + water vapor parcel. From a macroscopic point of view, one usually refers to "material entropy production" as the entropy produced by the geophysical fluids in the climate system, which are not related to the properties of the radiative fields, but rather to the irreversible processes related to the motion of these fluids. Mainly, this has to do with phase changes and water vapor diffusion, as outlined by Pauluis and Held (2002). Lucarini (2009) underlined the link between entropy production and efficiency of the climate engine, which were then used to understand climatic tipping points, and, in particular, the snowball/warm Earth critical transition (Lucarini et al., 2010b), to define a wider class of climate response metrics (Lucarini et al., 2010a), and to study planetary circulation regimes (Boschi et al., 2013). A constraint has also been proposed to the entropy production of the atmospheric heat engine, given by the emerging importance of non-viscous processes in a warming climate (Laliberté et al., 2015).

Given the multiscale properties of the climate system, accurate energy and entropy budgets are affected by subgrid-scale parametrizations (see also Kleidon and Lorenz (2004); Kunz et al. (2008)). These and the discretization of the numerical




scheme are generally problematic in terms of conservation principles (Gassmann and Herzog, 2015), and can eventually lead to macroscopic model drifts (Mauritsen et al., 2012; Gupta et al., 2013; Hobbs et al., 2016; Hourdin et al., 2017). See Lucarini and Fraedrich (2009) for a theoretical analysis in a simplified setting.

## 1.4 This Paper

We present here TheDiaTo v1.0, a new software for diagnosing the mentioned aspects of thermodynamics of the climate systems in a broad range of global-scale gridded datasets of the atmosphere. The diagnostic tool provides global metrics, allowing straightforward comparison of different products. These include:

- Top-of-Atmosphere, atmospheric and surface energy budgets;

- Total, atmospheric and oceanic meridional enthalpy transports;

- Water mass and latent energy budget;

- Strength of the LEC by means of kinetic energy dissipation conversion terms;

- Material entropy production with the direct and indirect methods;

The software is structured in terms of independent modules, so that the users are allowed to choose either all or part of the metrics according to their interest. A version of the tool has been developed as part of the ESMValTool community effort

(Eyring et al., 2016b) to provide a standardized set of diagnostics for the evaluation of Coupled Model Intercomparison Project Phase 6 (CMIP6) multi-model ensemble simulations (Eyring et al., 2016a). The current version of ESMValTool is v2.0a1. A new version is soon to be released including TheDiaTo v1.0. Therefore, our goal is to equip climate modellers and developers with tools for better understanding the strong and weak points of the models of interest. The final aim is to reduce the risk of a model having good outputs for quantities of common interest, such as surface temperature or precipitation, but for the wrong

dynamical reasons. Clearly, this is a first necessary step in the direction of creating a suite of model diagnostics composed of process-oriented metrics.

The dataset requirements are described (Section 2). We then clarify the methods used in each module (Section 3). In Section 4 an example is given of application with one 20-years-long model run. In section 5 an extensive evaluation is given of the evolution of various considered metrics under three different scenarios for which the CMIP5 (Taylor et al., 2012) provides a

multi-model ensemble. Summary and conclusions are given in Section 6.

## 2   Data and Software Requirements

The diagnostic tool consists of a set of independent modules, each, except the first one, being triggered by a switch decided by the user: energy budgets and enthalpy transports, hydrological cycle, Lorenz Energy Cycle, material entropy production with the direct or indirect method.





The software ingests all variables as gridded fields on a regular longitude-latitude grid covering the whole globe. For the LEC computation, 3D fields are required, stored in pressure levels at a daily or finer temporal resolution. If the model does not store data where the surface pressure is lower than the respective pressure levels, daily mean or higher resolution data of near-surface temperatures and horizontal velocity fields are also required for vertical interpolation. For all other computations,

2D fields are required as monthly means at Top-of-the-Atmosphere (TOA) and at the surface. Input data are given as separate NetCDF files. Variables are identified according to their variable names, that are required to comply to the Climate and Forecast (CF, http://cfconventions.org/Data/cf-documents/overview/article.pdf) and Climate Model Output Rewriter (CMOR, http://pcmdi.github.io/cmor-site/tables.html) standards. Datasets that are not complying with the CF-CMOR compliant names are reformatted through ESMValTool built-in preprocessing routines, if known (for more detail, refer to the dedicated report on

ESMValTool v2.0 Eyring et al. (2016b)). An overview of the required variables is provided in Table 1.

Energy budgets are computed from residuals of instantaneous radiative shortwave (SW) and longwave (LW) fluxes at TOA. At the surface they are combined with instantaneous turbulent latent and sensible heat fluxes (cfr. Section 3). The radiative fluxes (except the outgoing LW radiation, which is only upwelling) are composed of an upwelling and downwelling component, that are defined positive. The heat fluxes at the surface are positive upwards. Water mass and latent energy budgets are computed

from evaporation, rainfall and snowfall instantaneous fluxes. In case of recognized cumulative fields, these are converted to instantaneous by the ESMValTool preprocessor.

For the LEC module, 3D fields of velocity and temperatures, as well as near-surface temperature and horizontal velocity 2D fields are needed at the daily resolution. For the 3D fields, there is no specific constraint on the number of pressure levels, although the program has been tested on the standard pressure level vertical discretization used in CMIP5 outputs, consisting

on 17 levels from 1000 to 1 hPa. The program then subsets the troposphere between 900 and 100 hPa. LEC computation is performed on Fourier coefficients of the temperature and velocity fields. As further discussed later (Sect. 5.2), the vertical interpolation from near-surface fields to fill the empty fields in pressure levels, when necessary, is an inevitable source of uncertainty.

If explicitly requested by the user, the program is also able to perform computations of energy budgets, enthalpy transports and

the hydrological cycle on oceans and continents separately, provided a land-sea mask. This can either be in the form of land area fraction or of a binary mask.

The ESMValTool architecture is conceived as a library of Python script, whose latest version is available at www.doi.org/10.17874/ac8548f0315, where the dependency and installation requirements are also described. A stand-alone version of the software is maintained as well, relying on Python bindings for Climate Data Operators (CDO) (CDO, 2015).

## 3 Methods

### 3.1 Energy budgets and meridional enthalpy transports

Under the crucial assumptions of neglecting the heat content of liquid and solid water in the atmosphere, together with the heat associated with the phase transitions in the atmosphere and the effect of salinity and pressure in the oceans, we can write



the total specific energy per unit mass for the subdomains constituting the climate system as: $\epsilon = \mathbf{u}^2 + c_v T + \phi + Lq$ for the atmosphere, $\epsilon = \mathbf{u}^2 + c_w T + \phi$ for the ocean, $\epsilon = c_s T + \phi$ for solid earth and ice. Here $u$ is the velocity vector, $c_v$. $c_w$ and $c_s$ are the specific heat at constant volume of the atmospheric mix, water and the solid medium, respectively, $L$ is the latent heat of vaporization, $q$ the specific humidity and $\phi$ the gravitational potential. This leads to an equation for the evolution of the local

specific energy in the atmosphere as such (Peixoto and Oort, 1992):

$$\frac{\partial \rho \epsilon}{\partial t} = -\nabla \cdot (\mathbf{J_h} + \mathbf{R} + \mathbf{H_S} + \mathbf{H_L}) - \nabla(\tau \cdot \mathbf{u}), \tag{1}$$

where $\mathbf{J_h} = (\rho E + p)\mathbf{u}$ is the specific enthalpy transport, $\mathbf{R}$ is the net radiative flux, $\mathbf{H_L}$ and $\mathbf{H_S}$ are the turbulent sensible and latent heat fluxes respectively, whereas $\tau$ is the stress tensor. If we neglect the kinetic energy component, and vertically integrate Eq. 1, we can write an equation for the energy tendencies at each latitude, longitude and time for the whole climate

system, for the atmosphere, and for the surface below it:

$$\begin{cases} \dot{E}_t(\lambda, \phi) = S_t^\downarrow(\lambda, \phi) - S_t^\uparrow(\lambda, \phi) - L_t(\lambda, \phi) - \nabla \cdot J_t(\lambda, \phi) = R_t(\lambda, \phi) - \nabla \cdot J_t(\lambda, \phi) \\ \dot{E}_a(\lambda, \phi) = R_t(\lambda, \phi) - F_s(\lambda, \phi) - \nabla \cdot J_a(\lambda, \phi) = F_a(\lambda, \phi) - \nabla \cdot J_a(\lambda, \phi) \\ \dot{E}_s(\lambda, \phi) = S_s(\lambda, \phi) + L_s(\lambda, \phi) - H_S^\uparrow(\lambda, \phi) - H_L^\uparrow(\lambda, \phi) - \nabla \cdot J_o(\lambda, \phi) = F_s(\lambda, \phi) - \nabla \cdot J_o(\lambda, \phi) \end{cases} \tag{2}$$

where $\dot{E}_t$, $\dot{E}_a$, $\dot{E}_s$ denote the total, atmospheric and oceanic energy tendencies respectively. $S_s = S_s^\uparrow(\lambda, \phi) - S_s^\uparrow(\lambda, \phi)$, $L_s = L_s^\uparrow(\lambda, \phi) - L_s^\uparrow(\lambda, \phi)$ are the net SW and LW radiative fluxes at the surface, respectively, and $S_t^\downarrow, S_s^\downarrow, S_t^\uparrow$ and $S_s^\uparrow$ the TOA ($t$) and surface ($s$) upward ($\uparrow$) and downward ($\downarrow$) SW radiative fluxes respectively (and similarly for LW radiative fluxes, denoted by

$L$, provided that the downward LW flux at TOA is neglected). $R_t$, $F_a$ and $F_s$ are the total, atmospheric and surface net energy fluxes respectively. Net fluxes are defined as positive when there is a net energy input, negative when there is a net output. $J_t$, $J_a$ and $J_o$ are the meridional total, atmospheric and oceanic enthalpy transports. Oceanic transports are assumed to be related to surface EBs, because one assumes that long-term enthalpy transports through land are negligible (Trenberth and Solomon, 1994; Trenberth et al., 2001).

When globally averaged, Eq. 2 can be rewritten as (Lucarini et al., 2014):

$$\begin{cases} \dot{E}_t = R_t \\ \dot{E}_a = R_t - F_s = F_a \\ \dot{E}_s = F_s \end{cases} \tag{3}$$

Given the much smaller thermal inertia of the atmosphere, compared to the oceans, the TOA energy imbalance is expected to be transferred for the most part to the ocean interior in terms of ocean heat uptake, whereas $F_a$ is much smaller than $F_s$ (Levitus et al., 2012; Loeb et al., 2012; Smith et al., 2015). This is usually not found in climate models (e.g. Lucarini and

Ragone (2011); Loeb et al. (2015); Lembo et al. (2018)).

Under steady-state conditions, the tendency of the internal energy of the system vanishes when averaged over sufficiently long



timescales. We can thus zonally average Eqs. 2 and derive the long-term averages of the meridional enthalpy transports:

$$
\begin{cases}
T_t(\phi) = 2\pi \int_{\phi}^{\pi/2} a^2 \cos\phi' < \overline{R_t(\phi')} > d\phi' \\
T_a(\phi) = 2\pi \int_{\phi}^{\pi/2} a^2 \cos\phi' < \overline{F_a(\phi')} > d\phi' \\
T_o(\phi) = 2\pi \int_{\phi}^{\pi/2} a^2 \cos\phi' < \overline{F_s(\phi')} > d\phi'
\end{cases}
\tag{4}
$$

where $T_t$, $T_a$ and $T_o$ denote the steady-state total, atmospheric and oceanic meridional enthalpy transports, $a$ the Earth's radius, $<>$ long-term time averaging and overbars zonal-mean quantities. The stationary condition has to be achieved in the models

if the system is unforced. If the system is forced, for instance by means of transient greenhouse gas forcing, a correction is applied in order to prevent inconsistent cross-polar transports, as long as the weak non-stationarity condition holds (Carissimo et al., 1985), i.e. the latitudinal variability of the budgets is much larger than the global mean imbalance (cfr. Lucarini and Ragone (2011)). The correction to the energy fluxes is applied as:

$$
< \overline{B_x}(\phi) >_{co} = < \overline{B_x}(\phi) > - < \overline{B_x}(\phi) > /2\pi a
\tag{5}
$$

where $B_x$ refers to either $R_t$, $F_a$ or $F_s$.

Peak intensities and latitudinal locations are computed as metrics for such transports. Comparison among datasets with different resolutions is ensured by interpolating to a common grid.

## 3.2 Hydrological cycle

The atmospheric moisture budget is obtained from globally averaging precipitation and evaporation fluxes. The latter are

derived from surface latent heat fluxes as:

$$
E = \frac{H_L}{L_v}
\tag{6}
$$

where $L_v = 2.5008 \times 10^6 \ J/kg$ is the latent heat of evaporation (assumed to be constant). If $H_L$ is given in units of $W/m^2$, the implied evaporation flux is coherently given in units of $kg * m^{-2} * s^{-1}$. Rainfall fluxes are derived from total ($P$) and snowfall ($P_s$) precipitation as:

$$
P_r = P - P_s
\tag{7}
$$

Under the stationarity assumption, the equation for moisture budget is thus written as:

$$
\overline{E} - \overline{P} = \overline{E} - \overline{P_r} - \overline{P_s}
\tag{8}
$$

where overbars denote global averages.

For the latent energy budget in the atmosphere $R_L$, the following is used:

$$
\overline{R_L} = \overline{H_L} - L_v \overline{P_r} - (L_v + L_f) \overline{P_s}
\tag{9}
$$

where $L_f = 3.34 \times 10^5 \ J/kg$ is the latent heat of fusion (assumed to be constant). Unlike the water mass budget of Eq. 8, the latent energy budget is not closed, because it does not include the heat captured by snow fusion from the ground. Furthermore, the sublimation of ice is also not considered for sake of simplicity, despite it plays a significant role over the Arctic region.





### 3.3 The Lorenz Energy Cycle

The derivation of the atmospheric LEC in the diagnostic tool follows directly from the general framework proposed by Lorenz (1955), revised by Ulbrich and Speth (1991) in order to provide a separation among different scales of motion. The LEC allows to investigate the general circulation of the atmosphere by looking comprehensively at the energy exchanges between

eddies and zonal flow, at the conversion between available potential and kinetic energy, and at the dissipation due to viscous and mixing processes. Energy is injected in the reservoir of zonal APE through differential diabatic heating impacting both the zonal component (primarily) and the eddy component of the APE. The so-called baroclinic instability (mainly occurring in the mid-latitudinal baroclinic eddies at the synoptic scale) acts in two steps, by transforming zonal mean APE into eddy APE, which is then converted into eddy kinetic energy (KE) via a lowering of the atmospheric centre of mass. The eddy KE

is then converted into smaller scales through the kinetic energy cascade, eventually reaching the dissipative scale, where it is converted into frictional heating. A remaining part of the eddy KE is converted back into zonal mean flow through the so-called barotropic governor, ensuring the maintenance of the tropospheric and stratospheric jet streams (cfr. Li et al. (2007)).

The LEC is depicted diagrammatically in Figure 5, and the equations used to obtain the energy reservoirs and conversion terms are reported in the appendices A2-A3.

In the formulation proposed by Lorenz and widely adopted afterwards, one assumes that motions are quasi-hydrostatic. This assumption is correct as long as one considers sufficiently coarse-grained atmospheric fields. A detailed analysis of non-hydrostatic effects requires dealing with the exchange of available potential and kinetic energy taking place through accelerated vertical motions. Furthermore, the energetics refer to the dry atmosphere (cfr. Appendix A2-A3). In order to take moisture into account, one would have to replace the temperatures with virtual temperatures. The impact of such an assumption on the

estimate of the LEC is left for future study.

Following the arguments by Lucarini et al. (2014), we notice that, under non-stationary state conditions, the requirement for internal energy conservation holds (cfr. Eqs. 4), and the stationarity condition implies that APE and KE tendencies both vanish. Given that the tendency in KE is a balance of the APE-KE overall conversion (or in other words the mechanical work exerted by the LEC) and the dissipation of KE, we can write:

$$<W> = <D> = \int_{\Omega} \rho \kappa^2 d\mathbf{\Omega}', \tag{10}$$

where $W$ and $D$ denote the LEC work and the dissipation of KE respectively, $\kappa^2$ is the specific kinetic energy dissipation rate, $\rho$ is the atmospheric density and $\mathbf{\Omega}$ the volume of the atmosphere. This justifies the fact that the intensity of the LEC is used to obtain the kinetic energy dissipation of the atmosphere. Other terms, i.e. the generation/dissipation of APE and KE, are computed as residuals of the conversion terms at each reservoir.

### 30 3.4 Material entropy production

The total entropy production in the climate system is given by two qualitatively different kinds of processes: on one hand, the irreversible thermalization of photons emitted in the Sun's corona at the much lower Earth's surface and atmospheric





temperature, on the other hand the irreversible processes responsible for mixing and diffusion in the fluids and in the active soil of the climate system. The former accounts for roughly 95% of the total entropy production, the latter is the material entropy production (MEP) in a proper sense, and is the quantity of most interest in climate science, because it involves the dynamics of the atmosphere and its interaction with the Earth's surface (see discussion in Kleidon (2009); Lucarini (2009); Lucarini et al.

5 (2011, 2014))

In the long-term mean, assuming that the system is in statistically steady state condition, one can write an equation for the entropy rate of change in the system (Lucarini and Ragone, 2011; Lucarini and Pascale, 2014) as:

$$
\int_{\Omega} d\mathbf{\Omega}' \overline{\left(\frac{\dot{q}_{rad}}{T}\right)} + \overline{\dot{S}_{mat}} = 0 \tag{11}
$$

with $\dot{q}_{rad}$ denoting the local net radiative heating and $\dot{S}_{mat}$ the global material entropy production. From Eq. 11 it is evident that

computing the entropy rate of change as a consequence of radiative heating is equivalent to compute the sum of the entropy production from all the irreversible processes, both viscous (such as the energy dissipation) and non-viscous (such as the hydrological cycle). These two equivalent methods are here referred to as "indirect method" and "direct method", respectively. As long as the volume integral in the first lhs member of Eq. 11 is performed on the whole atmospheric domain, the two methods are exactly equivalent (for sake of simplicity we assume that this amounts to the MEP of the climate itself, since the

oceanic contribution to the MEP is neglected, accounting for about 2% of the budget, as found by Pascale et al. (2011)).

### 3.4.1 The direct method

The MEP computation with the direct method involves taking into account the viscous processes related to energy cascades toward the dissipative scales, non-viscous processes related to sensible heat fluxes (i.e. heat diffusion in the boundary layer, mainly dry air convection (Kleidon, 2009)) and to the hydrological cycle (such as evaporation in unsaturated air, condensation

in supersaturated air, release of gravitational potential energy due to the fall of the droplet and melting of solid phase water at the ground). Generally speaking, we can write the MEP equation as:

$$
\dot{S}_{mat} = \frac{\kappa^2}{T_v} + \frac{\phi}{T_\phi}, \tag{12}
$$

where $T_v$ and $T_\phi$ denote here the temperatures at which the respective processes occur, $\phi$ indicates local absorption or emission of heat that is neither radiative nor related to viscous dissipation. In other words, $\phi$ accounts for all the non-viscous irreversible

processes.

Overall, the hydrological cycle accounts for about $35 \, mW * m^{-2} * K^{-1}$, the sensible heat diffusion for about $2 \, mW * m^{-2} * K^{-1}$ and the viscous processes for about 6 to $14 \, mW * m^{-2} * K^{-1}$ (cfr. Fraedrich and Lunkeit (2008); Pascale et al. (2011)).

Dealing with direct computation of MEP in climate models has a number of additional implications. The mentioned irreversible processes are dealt with in climate models through the usage of subgrid-scale parametrizations. The fact that they are energy

conserving and entropy consistent is far from being trivial (Gassmann, 2013). Further, the numerical scheme adds spurious entropy sources, such as numerical advection and hyperdiffusion, as thoroughly addressed in an intermediate-complexity model





by Pascale et al. (2011) and focusing on the dynamical core of a state-of-the-art model by Gassmann and Herzog (2015). The relevance of these non-negligible numerically driven components, and the way to address them, is strictly model-dependent and a diagnostic tool that is meant to analyze a potentially diverse ensemble of datasets must come to terms with that limitation. In order to get to our expression for the direct MEP, let us first explicitly write the non-viscous terms in Eq. 12:

$$\dot{S}_{mat} = \int\limits_{\mathbf{\Omega}} d\mathbf{\Omega'} \frac{\kappa_s^2}{T} - \int\limits_{\Omega} d\mathbf{\Omega'} \frac{\nabla \cdot \mathbf{h}_S}{T} + \dot{S}_{hyd}, \quad (13)$$

where $\nabla \cdot \mathbf{h}_S$ denotes the heat diffusion through sensible heat fluxes, and $\dot{S}_{hyd}$ the aggregated MEP related to the hydrological cycle. The specific kinetic energy dissipation rate is here denoted by "s", in order to emphasize that it is now an estimate of such a term, as discussed later in this section. As argued by Lucarini and Pascale (2014), there is not easy way to account for the term related to the hydrological cycle. Pauluis and Held (2002); Pauluis (2011) find that the water mass in the atmosphere

can be thought as a "passive tracer", conveying heat until a phase change, i.e. an irreversible diabatic process, allows it to exchange heat with the surrounding, producing material entropy. This means that the atmospheric particle has to be separated in its dry component and the water mass components in its various phases, and the phase changes have to be evaluated, in order to address for the associated MEP (cfr. Pauluis and Held (2002); Raymond (2013)). At steady-state conditions an indirect estimate of $\dot{S}_{hyd}$ is provided as:

$$\dot{S}_{hyd} = \int\limits_{\mathbf{\Omega_V}} d\mathbf{\Omega_V} \rho_w \frac{\nabla \cdot \mathbf{h}_L}{T}, \quad (14)$$

where $\nabla \cdot \mathbf{h}_L$ denotes the heat exchange between the water mass particle and the surroundings during its phase transition.

In order to express Eq. 13 in terms climate model outputs, we make some additional assumptions, following the approach by Fraedrich and Lunkeit (2008). First, we assume that the energy dissipation by friction occurs mainly next to the surface, then we define an operating temperature $T_d$ as an average of surface ($T_s$) and near-surface temperatures ($T_{2m}$). Secondly, we

estimate the heat diffusion term from sensible turbulent heat fluxes at the surface $H_S$, transporting heat between the Earth's surface (having temperature $T_s$) and the boundary layer (having temperature $T_{BL}$, whose derivation is described below). We then consider the phase exchanges of water mass components as follows:

– evaporation at working temperature $T_s$ (indirectly derived from latent turbulent heat fluxes at the surface $H_L$);

– rain droplet formation through condensation at working temperature $T_C$, a characteristic temperature of the cloud;

– snow droplet formation through condensation+solidification at working temperature $T_C$, a characteristic temperature of the cloud;

– snow melting at the ground at working temperature $T_s$;



Following from Eq. 4 in Lucarini et al. (2011) we can thus rewrite Eq. 13 as:

$$\dot{\Sigma}_{mat} = \int\limits_{A} \frac{\kappa_s^2}{T_d} dA' - \int\limits_{A} H_S \left( \frac{1}{T_s} - \frac{1}{T_{BL}} \right) dA' - \int\limits_{A} \frac{L_v E}{T_s} dA' + \int\limits_{A_r} \left( \frac{L_v P_r}{T_C} + g \frac{P_r h_{ct}}{T_p} \right) dA'_r$$

$$+ \int\limits_{A_s} \left( \frac{L_s P_s}{T_C} + g \frac{P_s h_{ct}}{T_p} \right) dA'_s - \int\limits_{A_s} \frac{L_s P_s}{T_s} dA'_s \qquad (15)$$

Since the latent and sensible heat, rainfall and snowfall precipitation fluxes are given in model outputs as 2D fields, the volume integrals in Eq. 14 reduce to area integrals, with the domain $A$ denoting the Earth's surface and the subdomains $A_r$ and $A_s$

denoting the regions where rainfall and snowfall precipitation occur, respectively. The phase changes associated with snowfall and rainfall precipitation (4th and 5th rhs integrals) are accompanied by a term accounting for the potential to kinetic energy conversion of the falling droplet (with $g$ denoting the gravity acceleration and $h_{ct}$ the distance covered by the droplet). One may notice that, in principle, this is a viscous term, since the kinetic energy of the droplet is eventually dissipated into heat at the ground.

The first rhs member of Eq. 15 involves the specific kinetic energy dissipation rate ($\kappa_s^2$). There is no straightforward way to describe this quantity in climate models. Pascale et al. (2011) found that, besides the major role of the vertical momentum diffusion in terms of frictional stress in the boundary layer, the gravity wave drag and unphysical processes, such as horizontal momentum diffusion (hyperdiffusion), have also a non-negligible role. Each model accounts for these quantities in a different way. Generally, the frictional term is also not present in climate model outputs, thus has to be indirectly estimated from near-

surface velocity fields. In order to do so, one has to know the value of the drag coefficient, which is different in every model. In order to tackle this problem, also considering that the kinetic energy dissipation term overall contributes to less than 10% of the overall MEP, we compute it indirectly, obtaining the kinetic energy dissipation from the intensity of the LEC (cfr. Eq. 10). The boundary layer temperature $T_{BL}$ is not usually provided as climate model output, nor the boundary layer thickness is known a-priori. Some manipulations are thus needed. We start from the definition of a bulk Richardson number:

$$Ri_b = \frac{g}{\theta_{v0}} \frac{(\theta_{vz} - \theta_{v0}) z}{u_z^2 + v_z^2} \qquad (16)$$

where $g = 9.81\ m*s^{-1}$ is the gravity acceleration, $\theta_{v0}$ and $\theta_{vz}$ are the virtual potential temperatures at the surface and at level $z$, $u_z$ and $v_z$ are the zonal and meridional components of the horizontal wind at height z (assumed to be equal to the usually provided near-surface horizontal velocity fields). A critical Richardson number ($Ri_{bc}$) is defined as the value of the Richardson number at the top of the boundary layer. Its value depends on the nature of the local boundary layer (stable or unstable). In

order to distinguish among the stable and unstable boundary layers, a condition on the magnitude of the sensible heat fluxes is imposed (Zhang et al., 2014), so that where $H_S$ is lower than 0.75 $W/m^2$, a stable boundary layer is assumed ($Ri_{bc} = 0.39$; boundary layer height $z_{BL}$: 300 m), otherwise an unstable boundary layer is assumed ($Ri_{bc} = 0.28$; boundary layer height $z_{BL}$: 1000 m). For sake of simplicity, we approximate the virtual potential temperature as the dry potential temperature, so that the conversion from temperature to potential temperature and vice versa is given by the basic formula:

$$\theta = T \left( \frac{p}{p_0} \right)^{R_d/c_p} \qquad (17)$$





where $R_d = 287.0\ J*kg^{-1}*K^{-1}$ the gas constant for dry air and $c_p = 1003.5\ J*kg^{-1}*K^{-1}$ is the specific heat of the atmosphere at constant pressure. We can thus obtain $T_{BL}$ by solving Eq. 16 for $\theta_{z_{BL}}$, imposing $Ri_{bc}$ and $z_{BL}$ as mentioned, retrieving the value of $p$ at $z_{BL}$ by means of the barometric equation:

$$p_z = p_s e^{-gz/R_d T_s} \tag{18}$$

where $p_s$ the surface air pressure and $z$ is in our case $z_{BL}$, then using it to obtain the temperature at the boundary layer top. Here we assume that the boundary layer is approximately isothermal.

In Eq. 15, $T_C$ is a temperature representative of the interior of the cloud from where the moist particle drops. In order to define that, we first retrieve the dewpoint temperature at the surface from the equation:

$$T_d = \frac{1}{1/T_0 - R_v/L_v \log(e/\alpha)} \tag{19}$$

where $T_0 = 273.15\ K$ is the reference melting temperature, $R_v = 461.51\ J*kg^{-1}*K^{-1}$ is the gas constant for water vapor, $e = \dfrac{q_s p_s}{q_s + R_d/R_v}$ is the water vapor pressure (where we have used $q_s$, i.e. the near-surface specific humidity), $\alpha = 610.77\ Pa$ is one of the empirical parameters of the Magnus-Teten formulas for saturation water pressure (cfr. Goff (1957); Buck (1981)). An empirical formula for the computation of the lifting condensation level (LCL) (Lawrence, 2005) can be then used:

$$h_{LCL} = 125\,(T_s - T_d) \tag{20}$$

If we assume that the moist particle is lifted following a dry adiabatic until it saturates at the LCL, the temperature at such level will be:

$$T_{LCL} = T_s - \Gamma_d h_{LCL} \tag{21}$$

This would be the temperature of the cloud bottom in convective conditions. We hereby assume that similar conditions apply to stratiform clouds. In order to obtain $T_C$, we average $T_{LCL}$ with the temperature of the cloud top, which is assumed to be

the emission temperature $T_E$ at TOA by inversion of the the Stephan-Boltzmann law with outgoing longwave (LW) radiation at TOA (i.e. $L_t$ in Sect. 3.1).

The potential energy of the droplets in Eq. 15 is retrieved assuming that the particle starts to fall from the cloud layer top ($h_{ct}$). This level is obtained by assuming that the saturated particle, after entering the cloud at the LCL, continues to be lifted in the cloud on a pseudo-adiabatic path. We thus firstly compute the pseudo-adiabatic lapse rate:

$$\Gamma_p = \Gamma_d \left(1 + \frac{L_v q_s}{R_d T_{LCL}}\right) * \left(1 + \frac{\epsilon L_v^2 q_s}{c_p R_d T_{LCL}^2}\right)^{-1} \tag{22}$$

where $\epsilon = 0.622$ is the molecular weight of water vapor/dry air ratio. Once the pseudo-adiabatic lapse rate is known, it is straightforward to compute the height of the cloud top combining it with the emission temperature. It can be observed that what we obtain is an upper constraint to the potential energy of the droplets, since we assume that the particle falls through the whole cloud layer, while the pseudo-adiabatic lapse rate assumes that water vapor gradually precipitates during the ascent. $T_p$





is finally obtained as an average between $T_C$ and $T_s$.

In conclusion, let us notice that the MEP budget provided in Eq. 15 is the most accurate estimate that can be obtained from largely available climate model outputs. Still, some processes related to intermediate phase transitions in the atmosphere and heat exchanges at the droplet surface during its coalescence/aggregation stage are not taken into account, because the

information provided in model outputs does not allow for that. These terms are potentially relevant, as stressed by Pauluis and Held (2002) and Raymond (2013). Furthermore, the contribution from potential energy of the droplet does not usually enters the energetics of a climate model, although it is not negligible. Finally, the MEP budget here introduced is focused on the atmosphere. Phase changes in the sea-ice domain provide a potentially large contribution to the overall MEP of the climate (e.g. Herbert et al. (2011)).

### 3.4.2   The indirect method

For the indirect formulation of the entropy budget, we express the entropy associated with radiative heat convergence in a simplified formulation, following Lucarini et al. (2011). This approach is formally equivalent to the one adopted in Bannon (2015); Bannon and Lee (2017), using the definition of control volume to describe the entropy of the material system, together with the radiation contained in it. Not entering into details of the derivation, we follow from Eq. 11 by identifying those

processes responsible for the entropy production through exchanges of radiative energy, as outlined by Ozawa et al. (2003); Fraedrich and Lunkeit (2008). For each process, we thus define an energy flux between two mediums with warmer and colder temperatures. The radiative heat exchange is thus carried on locally through vertical exchanges, and on a large scale, mainly through meridional exchanges. If we take SW and LW net fluxes at the surface and at TOA (i.e. the usual output for radiative fluxes in climate models) we can write:

$$\overline{\dot{S}^{mat}_{ind}} = \int_A \frac{\overline{S_s} + \overline{L_s}}{T_s} dA + \int_A \frac{\overline{S_t} - \overline{S_s}}{T_{A,SW}} dA + \int_A \frac{\overline{L_t} - \overline{L_s}}{T_{A,LW}} dA \tag{23}$$

where $S_t = S_t^{\downarrow} - S_t^{\uparrow}$ is the net SW radiative flux at TOA (cfr. Eq. 2), $A$ is the surface area of the atmosphere, and $T_s$, $T_{A,SW}$ and $T_{A,LW}$ are characteristic temperatures representative of the surface and the portion of atmosphere where LW and SW radiative heat exchanges occur, respectively. Analogously to Eq. 15 the volume integral in Eq. 11 is transformed into an area integral, considering that the radiative fluxes occur at the surface of the domain and using the Gauss theorem (Lucarini et al.,

2011). This formulation is still an exact expression for the atmospheric MEP, as long as one is able to define SW and LW working temperatures $T_{A,SW}$ and $T_{A,LW}$ (cfr. Bannon and Lee (2017) for a discussion on this crucial issue). We follow again the arguments by Lucarini et al. (2011), rewriting Eq. 23 under the assumption that $T_{A,SW} \approx T_{A,LW} \approx T_E$:

$$\overline{\dot{\Sigma}^{mat}_{ind}} = \int_A \left( \overline{S_s} + \overline{L_s} \right) \left( \frac{1}{T_s} - \frac{1}{T_E} \right) dA + \int_A \frac{\overline{S_t} + \overline{L_t}}{T_E} dA = \overline{\Sigma_{ver}} + \overline{\Sigma_{hor}} \tag{24}$$

This decisive assumption is based on the fact that most of SW and LW radiation is absorbed and emitted in the atmosphere

through water vapor into the troposphere (Kiehl and Trenberth, 1997).

As already stressed by Lucarini et al. (2011), the material entropy budget described in Eq. 24 consists of two terms. The first



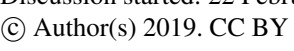

term denotes the vertical energy transport between a reservoir at temperature $T_s$ (the surface) and another at temperature $T_E$ (the TOA). For this reason it is referred to as "vertical material entropy production $(\overline{\Sigma_{mat}^{ver}})$. This term is positive almost everywhere and accounts for the vertical transport of warm air from the surface, mainly embedded in moist convection. The second term denotes the horizontal energy transport from a warm reservoir at lower latitudes to a cold reservoir at higher latitudes. This

is referred to as "horizontal material entropy production" $(\overline{\Sigma_{mat}^{hor}})$ and is associated with the annual mean meridional enthalpy transport setting the ground for the mean meridional circulation (Peixoto and Oort, 1992). One may notice that, while the first term accounts for a local entropy budget, the second has to be rather considered a horizontal advection of entropy, and should be meaningfully considered only in its global integral. Both terms are positive, the first one because the atmosphere is heated from below, the second because the temperature is lower where the heat is transported.

Let us finally consider that both the direct and the indirect methods contain crucial approximations. On the other hand, the 2-layer assumption is critical, as shown by Lucarini and Pascale (2014), finding that the coarse graining of post-processed model data always reduces the estimated MEP, both with the direct and indirect methods, with the indirect method particularly affected by vertical coarse graining. We expect that the indirect method leads to an overestimation of MEP compared to approximate estimates by Ambaum (2010)), mainly because of the vertical entropy production, as already seen in Lucarini et al. (2014). The

impact of considering the 3D radiative fluxes in Eq. 11 is under investigation with a specific intermediate-complexity model, and the outcomes will be the topic of a successive work.

## 4 Results from a CMIP5 model

Figure 1 about here

A 20-years subset of a CMIP5 model (CanESM2) simulation under pre-industrial conditions is here analysed, in order to describe the capabilities of the diagnostic tool. The datasets are retrieved from the Earth System Grid Federation (ESGF) node at

Deutsches Klimarechenzentrum (DKRZ). The run used here for the analysis is the one denoted by "r1i1p1". Of the 995-years (2015-3010) run, we take the 2441-2460 period.

Figure 1 shows the horizontal distribution of annual mean $R_t$, $F_a$ and $F_s$. The TOA energy budget $(R_t)$ is relatively smooth and zonally symmetric, with an area of net energy gain over the tropics and the ocean's subtropics, energy loss elsewhere. A maximum is found over the Eastern Indian - Western Pacific warm pool, where the Indian Monsoons develop and the emis-

sion temperature is the lowest, due to the presence of deep convection. Interestingly, this pattern is somewhat opposed by the negative values of the TOA EB at similar latitudes over the Sahara, where the highest near-surface temperatures are found. In other words, the warm and dry conditions characterizing deserted subtropical regions determine the highest thermal emission, largely exceeding the solar input, thus leading to a net energy loss. The surface energy budget $(F_s)$ is almost vanishing over the continents, given their small thermal inertia of the land surface. The largest absolute values are found in proximity of the

main surface ocean currents. They are mainly negative in coincidence of the western boundary currents (Gulf Stream, Kuroshio current, Agulhas current), where the ocean's surface transfers heat to the atmosphere. They are negative over the Humboldt Current, extending deep into the Equatorial Counter Current, and to a lesser extent in proximity of the Antarctic Circumpolar





Current. The atmospheric energy budget is clearly a local balance of the TOA and surface energy budget distribution, the most remarkable feature being the minimum in coincidence of the Equatorial Pacific.

Figure 2 about here

The meridional sections of climatological annual mean total, atmospheric and oceanic northward meridional enthalpy trans-
ports are shown in Figure 2. The figure layout follows from the classical approach on meridional transports implied from budgets and their residuals (e.g. Trenberth et al. (2001); Lucarini and Ragone (2011); Lembo et al. (2016)). The transports are vanishing at the Poles by definition, since Carissimo et al. (1985) correction (cfr. Equation 5) is applied, accounting for the effect inconsistent model energy biases. The atmospheric transport is slightly asymmetric, being stronger in the SH than in the NH. This is closely related to the asymmetry in the mean meridional circulation, being the latitude where the transport
vanishes coincident with the annual mean position of the Inter-tropical Convergence Zone (ITCZ), slightly north of the Equator (Schneider et al., 2014; Adam et al., 2016). The atmospheric transport peaks at about 40 degrees of latitude in both hemispheres, slightly more poleward in the NH. The peaks mark the regions where baroclinic eddies are mainly responsible of transporting energy poleward, and the zonal mean divergence of moist static energy switches sign (positive toward the Equator, negative toward the Poles; Loeb et al. (2015)). The oceanic transport is much less homogeneous than the atmosphere. The two peaks are
located in coincidence of the subtropical and midlatitudinal gyres, the second being smaller than the first. At mid-latitudes of the Southern Hemisphere a relative maximum is found, in some models (cfr. Figure 8) even denoting a counter-transport from the South Pole toward the Equator. This is a critical issue, evidencing that the reproduction of the Southern Ocean circumpolar current is a major source of uncertainty in climate models (Trenberth and Fasullo, 2010).

Figure 3 about here

Figure 3 shows the relation between atmospheric and oceanic peak magnitudes. CanESM2 exhibits a clear relation between the two quantities in the SH. A stronger oceanic peak corresponds to a weaker atmospheric peak, whereas the relation is less clear in the NH. The anticorrelation of oceanic and atmospheric peaks suggests that the compensation mechanism theoretised by Stone (1978) is well reproduced by the model, confirming that the shape of the total meridional heat transports is constrained by geometric and astronomical factors. This is far from being a trivial argument, since changes in the meridional planetary albedo
differences can deeply affect the total enthalpy transports (Enderton and Marshall, 2009), as well as the ocean-atmosphere partitioning (Rose and Ferreira, 2013). These plots facilitate an evaluation of these arguments in different models and under different scenarios and forcings.

Figure 4 about here

Figure 4 shows the annual mean horizontal distribution of water vapor in the atmosphere and its zonal mean meridional north-
ward transport. This point of view highlights the sources and sinks of humidity in the atmosphere, evidencing that most of the exchanges are effected over the oceans. The water mass (and similarly the latent heat) budget is relatively weak over most of the continents with significant regional exceptions, notably the Amazon, Bengal and Indonesia, as well as parts of the western coast of the American continent. The water mass budget over these land areas is mainly negative, denoting an excess of precipitation with respect to evaporation. The zonal mean water mass transport (Fig. 4(b)) is mainly poleward, except in the SH (30S - Eq)
and the NH tropics (10N - 30N), essentially diverging humidity in both directions from both hemispheres oceanic subtropics.





Clearly, water mass (and similarly latent energy, not shown) is primarily advected toward the regions of deep convection, i.e. the ITCZ (slightly north of the Equator), and secondarily toward both hemispheres extra-tropics, where moisture is provided for the baroclinic eddies (cfr. Cohen et al. (2000)).

Figure 5 about here

The Lorenz Energy Cycle for one year of CanESM2 model run (Figure 5) shows how the energetics of atmospheric dynamics are dealt with in the diagnostic tool. The reservoirs of Available Potential Energy (APE) and Kinetic Energy (KE) are shown in the blue boxes, separately accounting for the zonal mean, for stationary eddies (eddies in the time averaged circulation) and for transient eddies (departure from zonal and time mean). The spectral approach also allows to consider the partition between planetary wavenumbers, synoptic wavenumbers and higher order eddies, although it is not shown here.

Most of the energy is stored in the form of APE in the zonal mean flux, and to a lesser extent in the zonal mean kinetic energy. The zonal mean APE is almost instantly converted into eddy potential energy (mainly through meridional advection of sensible heat) and then into eddy kinetic energy (through vertical motions in eddies) by means of mid-latitudinal baroclinic instability, so the two conversion terms are not surprisingly qualitatively similar. We also notice that the edddy APE and KE reservoirs have similar magnitudes. As argued before (Li et al., 2007), this is a consequence of the tight relation between temperature

perturbations to the zonal mean meridional profile and the eddy synoptic activity. CanESM2 (and other models as well, not shown) agrees well with observational-based datasets on the fact that the stationary eddies play a non-negligible role in the baroclinic-barotropic energy conversion (cfr. Ulbrich and Speth (1991)). As for the KE, the transient eddy reservoirs are about half of the zonal mean, with the stationary eddy playing a more marginal role. The barotropic conversion acts mainly by converting eddy into zonal mean KE (i.e. restoring the jet stream), but in part also converting APE into KE (or vice versa) in the

zonal mean flow.

Compared to Reanalysis datasets (e.g. Ulbrich and Speth (1991); Li et al. (2007); Kim and Kim (2013)), our approach features more energy stored in the zonal mean reservoirs. This is consistent with previous findings (cfr. Boer and Lambert (2008); Marques et al. (2011)), mentioning as possible reason for that the well-known cold Pole bias and the consequently excessive speed of the jet stream. Besides the fact that pre-industrial conditions feature different conditions than the present-day, as shown in

Table 7, another explanation is that, unlike previous results from Reanalysis, we only consider the tropospheric part of the Lorenz Energy Cycle. The conversions of APE to and from stationary eddies also diverge from Reanalyses (Kim and Kim, 2013), although the overall baroclinic conversion is consistent with them. For model inter-comparison in the next Section, we will then consider the sum of the stationary and transient eddies as a single eddy component. The KE-APE conversion in the zonal mean flow is problematic, with CanESM2 having an opposite sign as Reanalyses, although also these observational-

based products appear to have inconsistencies (Kim and Kim, 2013).

Figure 6 about here

Results obtained from the indirect method for MEP with CanESM2 model are shown in Figure 6, which shows the climatological annual mean maps of the vertical (a) and horizontal components (b). Two different color maps have been used, in order to emphasize that, despite the vertical component has smaller maximum values than the horizontal component, it is positive

almost everywhere (cfr. Lucarini et al. (2011)). As mentioned in Sect. 3.4.2, the local value of the horizontal component is not



meaningful per se, and this component shall only be addressed globally. Figure 6b is meant to describe an entropy flux divergence from the Tropics, particularly the Indian-Pacific warm pool, toward the high latitudes, roughly reflecting the atmospheric meridional enthalpy transport described in Fig. 2. This allows to provide a link between entropy production and the meridional enthalpy transports, with the null isentrope delimiting the areas of enthalpy divergence from those of enthalpy convergence
(cfr. Loeb et al. (2015)). The vertical entropy production features its highest values where the evaporation is most intense (cfr. Figure 4(a)). On the contrary, lowest values are found over the continents and the regions of subsidence in the atmospheric meridional circulation. In other words, the vertical component is indicative of a local budget, in which atmospheric columns are weakly coupled with each other and the mixing occurs on the vertical (Lucarini and Pascale, 2014).

## 5   Multi-model inter-comparison and changes across different scenarios

We now focus on comparing a 7-members multi-model ensemble from CMIP5 Project under three different scenarios: "piControl" (piC), denoting pre-industrial conditions, "historical" (hist), i.e. a realistic forcing evolution for the 1870-2005 period, and "rcp8.5", representing the 2005-2100 evolution of GHG forcing under business-as-usual emission scenario (in other words, a 8.5 $W/m^2$ forcing by the end of the 21th Century). For the "piC scenario, 20-years periods, not necessarily overlapping, have been considered for each model, for hist scenario the 1981-2000 period, for rcp8.5 the 2081-2100 period. The choice of the
7 models and the 20 years periods are motivated by the availability of the CMIP5 model outputs for the three experiments on the DKRZ ESG node. It can be said that this time length reflects the typical range for decadal climate predictions, as indicated in the Intergovernmental Panel on Climate Change (2013) report, so it well fits our aim to describe the mean state and inter-annual variability of the climate system with different conditions. A summary of the main global metrics described in Sect. 3 is reported in Tables 2-4.

20                                           Table 2 about here

Table 3 about here

Table 4 about here

Figure 7 about here

Figure 8 about here

25                                           Figure 9 about here

Figure 10 about here

### 5.1   Energy and water mass budgets, meridional enthalpy transports

As shown in the first two columns of Table 2, there is a large bias on the TOA energy budget under unforced piC conditions. Such bias sums up with the bias on atmospheric energy budget (not shown, see Eq. 3) resulting in the bias on surface energy
budget estimates (the multi-model mean value for $R_t$ being 0.21 $W/m^2$, for $F_s$ 0.73 $W/m^2$). Some clear outliers are found, having either negative (BNU) or positive (MIR-C) values. This bias is the signature of the well-known model drift in climate models (Gupta et al., 2013). The fact that the bias is larger on surface budgets is explained by the fact that (cfr. Mauritsen et al.





(2012); Hourdin et al. (2017)) models are generally tuned in order to achieve vanishing TOA budgets, whereas surface energy budgets are in most cases left as free parameters. Panels a-c of Figure 7 emphasize how these biases are relevant with respect to the inter-annual variability of the budgets (computed as the standard deviation of the annual mean values). The atmospheric inter-annual variability is roughly one order of magnitude smaller (about $0.1\ W/m^2$) than the variability on TOA and surface

budgets, emphasizing that the changes in the overall energy imbalance are transferred to a large extent into the ocean (cfr. also Fig. 7d). The model agreement on net energy fluxes is similar at the surface and at TOA, except for two models (BNU and MIR5) exhibiting very large biases in the atmosphere, which then reflect in TOA biases. A closer look on the correlation between surface and atmospheric biases suggests that they are quite uncorrelated with each other . The inter-annual variability is roughly the same order of magnitude as the biases, both in the atmosphere and at the surface/TOA (about $0.20$-$0.25\ W/m^2$).

As a consequence of a transient GHG forcing, the TOA imbalance is found to increase (cfr. Tables 3 and 4). By the end of the historical period, most models agree on showing a positive imbalance (with respect to unforced biased conditions), ranging between $0.2$ and $0.7\ W/m^2$, although still in the range of the inter-annual variability (cfr. Figure 7a). The imbalance is much stronger by the end of the RCP8.5 period, peaking at $2.8\ W/m^2$ (net of the bias, i.e. the value of the imbalance in piC scenario) in MIR-C. The surface imbalance appears to increase consistently with the TOA imbalance.

Figure 9 shows the difference in the meridional latent heat transport between the Equator and 10N. As mentioned in Sect. 4 (cfr. Fig. 4(b)), this is a measure of the moisture convergence toward the ITCZ. There is quite large uncertainty on its value in piC scenario, ranging between $1$ and $3\ PW$. In all models the convergence is found to increase by up to $1\ PW$ between piC and RCP8.5. Even though beyond the scope of this work, one may argue that this is a robust estimate of the intensity of the uplifting branch, and to some extent of the intensity of the Hadley circulation.

Table 5 denotes a discrepancy of several $W/m^2$ in the atmospheric budgets over ocean and land, with a positive imbalance over the former, negative over the latter. Such a well-known imbalance (see Wild et al. (2015) for a review on the model perspective) is key to understand the models capability to reproduce the hydrological cycle, which can be ultimately seen as the convergence of latent energy (cfr. Eq. 9) from oceans (where water mass evaporates) toward the continents (where a large part of it precipitates). Atmospheric energy and latent heat asymmetries are compared in the first two columns of Table 5.

Models that are relatively well balanced (Can2, IPSL-M, MPI-LR and MPI-MR) also feature relatively similar atmospheric and latent heat asymmetries. These asymmetries are translated into land-ocean transports, multiplying ocean and land fluxes by their respective surface area. The two transports are obviously required to equal each other, net of the sign (see Table 6). A reference estimate for such a transport amounts to $2.8\ PW$ (Wild et al., 2015). Only few models are found to stick to this basic energy conservation rule (with Can2, IPSL-M, MPI-LR and MPI-MR performing better than the others), featuring differences

up to $1.7\ PW$ for BNU. For the better performing models, it can be noticed that a residual asymmetry holds, which is not attributable to the latent heat asymmetry (third column in Table 5). Those transports are directed from land toward the oceans and are interpreted as the land-ocean transports related to asymmetries in the sensible heat fluxes at the surface. The ocean-land latent energy transport is found to increase in RCP8.5 by $0.4 - 0.8\ PW$. This can be interpreted as an increase in the strength of the hydrological cycle, in line with previous findings (e.g. Levang and Schmitt (2015)). Looking at individual components

of the hydrological cycle, we find an increase both in evaporation over oceans and precipitation over land.





The mean meridional sections of total, atmospheric vs. oceanic enthalpy transports for each model are shown in Figure 8 for the piC conditions alone (cfr. Tables 2-4 for an overview of peak magnitude and position values). The choice of not showing hist and rcp85 is motivated by previous arguments on the insensitivity of enthalpy transports to the different forcing (in agreement with the theory by Stone (1978), confirmed by previous findings on CMIP models behaviors in disparate forcings representative

of nowadays climate or future emission scenarios, e.g. Lucarini et al. (2011); Lembo et al. (2018)). Models agree on the asymmetry in the atmospheric transport, being stronger in the Southern Hemisphere than in the Northern Hemisphere (see Sect. 3). A significant source of uncertainty is the location of the zero-crossing, which is in some cases very close to the Equator or even displaced in the SH. This is an important metric for the strength and shape of the mean meridional circulation. Compared to the atmosphere, the uncertainty on oceanic enthalpy transports is much larger, especially in the Southern Hemisphere, with

some models featuring a counter-transport toward the Equator. As already mentioned, a typical source of uncertainty here is the relative maximum in the Southern Hemisphere extra-tropics.

### 5.2 The intensity of the LEC and its components

Table 7 about here

We provide in Table 7 the values of the components for the 7 models in the three scenarios. The eleventh column in Tables 2-4 evidences that the intensity of the Lorenz Energy Cycle is to a large extent constant through the scenarios. Its value is comprised

between $2.2\,W/m^2$ (Can2) and $1.1\,W/m^2$ (MIR5 in rcp8.5 scenario). As mentioned in Sect. 3.3, a major source of uncertainty is given by the way fields are vertically discretized in pressure levels across the different models. Some models (BNU, Can2, MPI-LR, MPI-MR) internally interpolate the fields, in order to provide meaningful value in each gridpoint at each level. Others have no values where the surface pressure is lower than the respective pressure level. This normally occurs over Antarctica and mountainous regions (most notably Himalaya and the Rocky Mountains), where the surface pressure reaches values even lower

than 700 hPa. Since the Lorenz Energy Cycle is computed on spectral fields, the original gridpoint fields have to be continuous, and the gaps, if present, have to be filled with a vertical interpolation. In order to do so, daily mean near-surface fields of zonal and meridional velocities, and near-surface temperatures are used. Near-surface velocities replace the gaps, whereas near-surface temperatures are interpolated on the vertical, considering a vertical profile of temperature reconstructed from barometric equations (cfr. Sect. 3.4.1). Despite the retrieved velocity and temperature profiles are qualitatively comparable with

the internally interpolated models, the stationary eddy conversion terms are unreasonably weaker. This is not surprising, since the interpolation mostly affects mountain regions over the mid-latitudinal continents, i.e. the regions where the orografically driven stationary planetary waves are generated.

As mentioned, the intensity of the LEC is not really sensitive to the type of forcing. The result is partially in contrast with with a previous version of MPI-LR (Hernández-Deckers and von Storch, 2010) and Reanalyses (Pan et al., 2017), though the

changes/trends in the APE-KE conversion (i.e. the intensity of the LEC) for both studies are not very strong. Nevertheless, Table 7 evidences relevant changes in some of the storage components. Not entering into details of transient vs. stationary eddies partitioning, we notice that the zonal mean APE is largely decreased from piC to rcp8.5. This is compensated by an increase of similar magnitude in the zonal mean KE (and, to a lesser extent, of the eddy KE). In other words, in the wake of



an increasing GHG forcing, the share of kinetic energy to available potential energy is changed in favor of the first. The total (APE+KE) energy contained in all storage terms remains to a large extent stationary, in agreement with Pan et al. (2017). These results are consistent with what previously found by Hernández-Deckers and von Storch (2010) with a previous version of MPI-ESM-LR, and Veiga and Ambrizzi (2013) with a state-of-the-art version of MPI-ESM-MR. The zonal-mean APE reduction is

consistent with a reduction of the meridional temperature gradient, predominantly as a consequence of high latitude warming amplification (cfr. also Li et al. (2007)). The increase in zonal mean kinetic energy reflects a strengthening of the tropospheric mid-latitude jet stream (consistently with what previously found about the SH tropospheric jet, e.g. Wilcox et al. (2012)). Less clear is the impact of climate change on the mid-latitudinal eddy activity. The (slight) increase in eddy kinetic energy may reflect an increased mid-latitudinal baroclinic eddy activity in the Pacific and Atlantic storm tracks (despite large uncertainty

still holds on the extent to which models agree on such changes, cfr. Intergovernmental Panel on Climate Change (2013)). This, together with the raise of the tropical tropopause (mainly as a consequence of surface warming, cfr. Lin et al. (2017)), may contrast the expected decrease in baroclinic eddy activity associated with a smaller meridional temperature gradient. In other words, this approach allows to straightforwardly associate the different response of the models to the increasing GHG with key aspects of the general circulation of the atmosphere.

## 15  5.3  Material entropy production in the two methods

The components of the material entropy production in the indirect and direct methods are closely related to each other and provide further insight into the interpretation of water mass, energy budgets and LEC results.

First, let us consider Table 8, summarizing the main components of the material entropy production in the two methods. The most part of the material entropy production obtained with the direct method is associated with the hydrological cycle,

whereas the vertical component overcomes the horizontal component in the indirect method by an order of magnitude. We have already commented on this in Sect. 3.4.2 and 4. Here we notice that previous arguments about the changes in intensity of the hydrological cycle and of the atmospheric circulation are here confirmed. The entropy production increases with increasing GHG forcing in all models (except BNU, that we have already noticed being strongly water mass and energy unbalanced). The increase in the MEP related to the hydrological cycle ranges between $2.4 \, mW * m^{-2} \times K^{-1}$ (MIR5) and $4.7 \, mW * m^{-2} \times K^{-1}$

(IPSL-M) from piC to RCP8.5, amounting to about a 10% increase. The vertical component increase ranges between $5.6 \, mW * m^{-2} \times K^{-1}$ (MPI-MR) and $7.3 \, mW/Km^2$ (IPSL-M). Models showing larger increases in the hydrological-cycle-related MEP also feature stronger increase in the vertical component. In other words the vertical component, being a signature of MEP related to vertical updrift, mainly through deep tropical convective activity, is closely relate to the strength of the hydrological cycle.

Table 9 provides more insight into the components of the MEP related to the hydrological cycle (cfr. Equation 15). We notice here that the two main changes are on the one hand the reduction in MEP due to less snowfall precipitation ($S_s$). This reduction ranges between $3 \, mW * m^{-2} \times K^{-1}$ (MIR5) and $8.5 \, mW/Km^2$ (BNU) from piC to rcp8.5, to which a reduction of less than $1 \, mW * m^{-2} \times K^{-1}$ must be added as a consequence as less snow melting. On the other hand a large increase in MEP due to rainfall precipitation ($S_r$) is found, ranging between $10.7 \, mW * m^{-2} \times K^{-1}$ (MIR5) and $32.1 \, mW * m^{-2} \times K^{-1}$ (IPSL-M)





from piC to rcp8.5, generally overcoming the MEP reduction related to evaporation at the surface ($S_s$). This increase can be interpreted in different ways, either as an increase in water mass which is precipitated, or in terms as a lower working temperature for rain droplet formation ($T_C$). As a consequence of the water mass balance, the latent heat associated with evaporation and rainfall precipitation have to equal each other (net of the changes in latent heat related to snowfall precipitation

and the marginal contribution by snow melting at the ground). We thus attribute such an increase in $S_r$ to changes in $T_C$. This might also be indicative of larger rate of convective precipitation on stratiform precipitation, and might be investigated further. Concerning the other terms of the material entropy production, the one related to the sensible heat fluxes at the surface is slightly reduced, whereas the kinetic energy dissipation term is increased. Given that the LEC intensity, from which the kinetic energy dissipation has been derived, is to a large extent stationary across the scenario, such change is not related to the intensification

of the atmospheric circulation (as argued in Sect. 5.2), rather attributable to the near-surface warming. Finally, the decrease in the horizontal component is in line with the decrease in the APE terms of the LEC (especially the zonal mean term), denoting a weaker heat convergence toward the high latitudes (mainly as a consequence of high latitudes amplified warming).

As a whole, the entropy production is found to increase with increasing GHG forcing (see Tables 2-4), both with the indirect and the direct method. This is consistent with previous findings (Lucarini et al., 2014, 2011), pointing at the role of latent

heat release in convective processes, setting up the response of the climate system. The reduction of the meridional enthalpy transports is also consistent with previous comparisons between dry and moist entropy fluxes (Laliberté and Pauluis, 2010), suggesting a limiting role of the hydrological cycle to the efficiency of the atmospheric thermal engine (Laliberté et al., 2015).

### 5.4   Baroclinic efficiency and irreversibility

As a wrap-up of the various aspects touched in this section, we introduce two metrics , the so-called "baroclinic efficiency"
(Lucarini et al., 2011):

$$\eta = \frac{T_E^> - T_E^<}{T_E^>} \tag{25}$$

, where $T_E^>$ and $T_E^<$ are the emission temperatures averaged in the domains defined by TOA net energy gain and net energy loss (cfr. Figure 1), respectively, and the "degree of irreversibility" (Lucarini et al., 2011):

$$\alpha = \frac{\overline{\dot{S}_{dir}^{mat}} - \overline{\dot{S}_k^{mat}}}{\overline{\dot{S}_k^{mat}}} \tag{26}$$

, i.e. the rato of MEP from irreversible processes others than the kinetic energy dissipation to the MEP related to the kinetic energy dissipation alone. The first parameter accounts for the strength of the mean meridional circulation, driven by the differential thermal gradient. In other words, this is a measure of the dry entropy fluxes related to the heat to work conversion associated with the existence of the LEC, i.e. an upper limit to the efficiency of the atmospheric thermal engine. The second parameter accounts for the relevance of viscous dissipation compared to other non-viscous irreversible processes. This param-

eter is closely related to the Bejan number, which is widely use in thermodynamics for the study of heat transfers in a fluid (Awad, 2016).

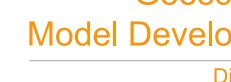
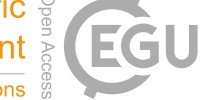


Table 10 shows the results from the three scenarios for the seven models that have been considered. The baroclinic efficiency ranges between 0.051 (MIR-C) and 0.066 (IPSL-M) in piC. It undergoes a clear reduction in the wake of increasing forcing, as a consequence of the already discussed reduction in the heat convergence at high latitudes. Remarkably, the last 20 years of hist do not seem to be significantly different from piC, contrary to rcp8.5. The irreversibility, in turn, is generally increased,

especially between hist and rcp8.5. As noted before, the hydrological cycle has an increasing importance in converting energy into the system in a warmer climate (cfr. also Laliberté and Pauluis (2010); Laliberté et al. (2015). This reflects a less efficient meridional enthalpy transport from low to high latitudes, and consequently in a larger irreversibility of the system, as already argued in Lucarini et al. (2010a, b).

## 5.5   Links among the metrics

Figure 10 brings up some of the metrics discussed up until now for the piC scenario, to describe to what extent they can be related with each other. The TOA and atmospheric budgets (Figure 10a) are here clearly related, with positive/negative outliers determined by positive/negative biases in atmospheric budgets. The other models cluster around vanishing atmospheric energy imbalances and slightly positive TOA imbalances, likely reflecting the oceanic model drift.

Figure 10b shows that the uncertainty on the value of baroclinic efficiency is much smaller (10%) than the one related to the

LEC (about 50%, even though the treatment of fields in pressure levels is here a critical issue, as discussed in Sect. 5.2). The two quantities are related through the meridional enthalpy transports (cfr. Boschi et al. (2013)), which in mid-latitudes are mainly effected through baroclinic eddies. As mentioned, it can also be noticed that the strength of the baroclinic conversions in the LEC are themselves a measure of the strength of the LEC. One might argue that the baroclinic efficiency peaks for certain values of the LEC intensity, but a larger ensemble would be necessary to prove or disprove such hypothesis.

Figure 10c shows the relation between horizontal and vertical components of the MEP computed with the indirect methods (cfr. Lucarini et al. (2014)). It can be noticed that larger/smaller values of the vertical components correspond to smaller/larger values of the horizontal component. Indeed, the overall MEP ranges between about 57 and 60 $mW * m^{-2} \times K^{-1}$, while the vertical component alone has a 4 $mW * m^{-2} \times K^{-1}$, suggesting that somewhat a compensation mechanism occurs between the vertical and horizontal component, i.e. between the local MEP on the vertical (especially where convection occurs) and the

convergence of heat towards the high latitudes.

Finally, Figure 10d shows the relation between MEP computed via the direct and indirect methods, respectively. Besides one outlier (BNU) all models have values of the direct method about 40 $mW * m^{-2} \times K^{-1}$, whereas through the indirect method they cluster about 57.5 $mW * m^{-2} \times K^{-1}$. Comparison with explicit computations by Pascale et al. (2011) suggest that the discrepancy is mainly attributable to an insufficient representation of the MEP related to the hydrological cycle (since some

intermediate phase change processes are not taken into account here) and to the kinetic energy dissipation (because unphysical processes cannot be accounted for here, given the different numerical schemes of the models, leading to up to 50% reduction of this term).

Summarizing, this analysis emphasizes that the distribution of different models can be quite different from being Gaussian.





The multi-model ensemble mean and variances here used are certainly useful criteria for assessing the model uncertainty, but they shall be carefully used, always paying attention to the behavior of each member of the ensemble.

# 6 Summary and Conclusions

We have here presented TheDiaTo v1.0, a diagnostic tool for the study of different aspects of the thermodynamics of the climate system, with a focus on the atmosphere. The goal of this diagnostic tool is to support the development, the evaluation, and the intercomparison of climate models, and to help the investigation of the properties of the climate in past, current, and projected future conditions. The diagnostic tool consists of few indipendent modules, accounting for: (1) the energy budgets and transports in the atmosphere, the oceans, and in the system as a whole, (2) the water mass and latent energy budgets and transports, (3) the Lorenz Energy Cycle (LEC), (4) the material entropy production with the direct or indirect methods. Global metrics are provided for immediate comparison among different datasets.

We provide some examples of practical use of the diagnostic tool. We have presented results obtained from a 20-years subset of CMIP5 model run under unforced pre-industrial conditions, and results from a 20-years multi-model ensemble in three different scenarios: unforced pre-industrial conditions (piC), the end of the historical period (hist) and the last 2 decades of the 21st Century with a business-as-usual scenario (rcp8.5). A summary of the metrics and of the comparisons between the results obtain across models and scenarios is given in Tables 2-4 and in Figure 10.

The energy and water mass budgets have been computed locally, but from this the transports are inferred, providing information about the global scale circulation. Similarly, the material entropy production has been decomposed in a component which essentially accounts for a local budget on the vertical, and another one which accounts for the global meridional enthalpy transport. In other words, the metrics here shown link the local features of the climate to the global energy and mass exchange, allowing for the evaluation of the global impact of localized changes.

We have shown how the tool can provide a comprehensive view of the dynamics of the climate system and its response to perturbations. On one hand, it is possible to evaluate the spatial distributions of model biases and their impacts, on the other hand we can interpret the change of the properties of the system with time in the reduced space defined by the considered metrics.

Apart from the specific - yet important - problem of loking into climate change scenarios, it is thus straightforward to use the diagnostic tool for the study of paleoclimate, investigation of tipping points, the study of the climate under varied astronomical factors, chemical composition of the atmosphere. One can envision adapting the model for the analysis of the properties of Earth-like exoplanets.

The requirement of flexibility, which allows the tool to work for a large class of models, inevitably leads to make some simplifying hypotheses on the system. The most relevant are the following: a) the quasi-steady-state assumption; b) the hydrostatic assumptions, as background to the LEC framework; c) the identification of the emission temperature as the characteristic temperature in the atmosphere, leading to the 2D formulation of the material entropy production with the indirect method. Other assumptions involve the analysis of the hydrological cycle, as the latent heat of evaporation and solidification have been as-





sumed as constant, even though their value depends on temperature and pressure. Further, it is worth noticing (Loeb et al., 2015) that the latent energy associated with snowfall melting over sea-ice ocean free regions is not accounted for in CMIP5 models. This accounts for about $0.1 - 0.5\ W/m^2$. Nevertheless, unlike the water mass budget the latent energy budget is not expected to be closed, since the surface melting is not taken into account (although it is considered for the entropy budget).

Thus far, we have pointed out that the thermodynamic point of view can be clearly linked to fundamental aspects of the atmospheric dynamics. We have linked the idea of a baroclinic heat engine (Barry et al., 2002) to the work carried in a Lorenz Energy Cycle (Lorenz, 1955), along the lines of what proposed by Lucarini et al. (2010a, 2014). A deeper insight into the energetics of the atmospheric dynamics would require an evaluation of the meridional mass streamfunctions. To this aim, an additional tool for the streamfunctions is being developed on moist and dry isentropes, rather than on isobaric coordinates,

following from Laliberté et al. (2015); Kjellsson (2015), in order to link the Lagrangian and the Eulerian point of view. The extent to which this idea can be pursued obviously depends on the availability of model outputs. Still, we have shown here how deeply is the hydrological cycle affected by changes in the mean state of the system, and such an isentropic approach allows for resolving the heat exchanges due to moist processes.

Another open issue is assessing the relevance of coarse graining for the results, not only on the material entropy production

(as discussed by Lucarini and Pascale (2014)), but also in terms of LEC and efficiency (when it comes to the type of vertcial discretization of the model). On one hand, the method is being tested with 3D fields for the radiative fluxes. On the other hand, the impact of changing the temporal, vertical and horizontal resolution is being assessed through a number of dedicated sensitivity studies with an intermediate complexity atmospheric model.

*Code and data availability.* The diagnostics are part of the ESMValTool community diagnostics (v2.0). The latest release of the tool is

available for download at www.doi.org/10.17874/ac8548f0315. CMIP5 data have been gathered from the ESGF node at DKRZ, publicly available upon registration https://esgf-data.dkrz.de/projects/esgf-dkrz/

.

## Appendix A: Sources, sinks and conversion terms of the Lorenz Energy Cycle

### A1    Symbols and Definitions

– $c_p$ = specific heat at constant pressure

– $g$ = gravity

– $p$ = pressure

– $r$ = Earth's radius

– $t$ = time

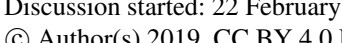



- $T$ = temperature

- $T_V$ = virtual temperature

- $u$ = zonal velocity

- $v$ = meridional velocity

– $\omega$ = vertical velocity

- $\gamma = -\dfrac{R}{p}\left(\dfrac{\partial}{\partial p}[\overline{T}] - \dfrac{\partial}{\partial p}\dfrac{[\overline{T}]}{p}\right)^{-1}$

- $\overline{x}$ = time mean of x

- $x'$ = deviation from time mean

- $x$ = global horizontal mean

– $[x]$ = zonal mean

- $x^*$ = deviation from zonal mean

## A2   Storage terms

- $ZPE = \dfrac{\gamma}{2g}\left([\overline{T}] - \{\overline{T}\}\right)^2$

- $EPE = \dfrac{\gamma}{2g}\left([\overline{T^{*2}}] + [\overline{T'^2}]\right)$

– $ZKE = \dfrac{1}{2g}\left([\overline{u}]^2 + [\overline{v}]^2\right)$

- $EKE = \dfrac{1}{2g}\left[\overline{u^{*2}} + \overline{v^{*2}} + \overline{u'^2} + \overline{v'^2}\right]$

## A3   Conversion terms

-

$$C_A = -\frac{\gamma}{g}\left\{\frac{\partial[\overline{T}]}{r\partial\phi}[\overline{v'T'} + \overline{v^*}\overline{T}^*] + [\overline{\omega'T'} + \overline{\omega^*}\overline{T}^*]\left(\frac{\partial}{\partial p}([\overline{T}] - \{\overline{T}\}) - \frac{R}{pc_p}([\overline{T}] - \{\overline{T}\})\right)\right\} +$$

$$+ \frac{\gamma}{g}\left\{+\overline{u'T'}^*\frac{1}{r\cos\phi}\frac{\partial\overline{T}^*}{\partial\lambda} + \overline{v'T'}^*\frac{\partial\overline{T}^*}{r\partial\phi}\right\}$$

– $C_Z = -\dfrac{R}{gp}\left([\overline{\omega}] - \{\overline{\omega}\}\right)\left([\overline{T_v}] - \{\overline{T_v}\}\right)$

- $C_E = -\dfrac{R}{gp}\left[\overline{\omega^*}\overline{T_v}^* + \overline{\omega'T'_v}\right]$





$$
\begin{aligned}
C_K = -\frac{1}{g} \Bigg\{ & \left( \frac{\partial[\overline{u}]}{r\partial\phi} + [\overline{u}]\frac{\tan\phi}{r} \right) [\overline{u}^*\overline{v}^* + \overline{u'v'}] + \frac{\partial[\overline{v}]}{r\partial\phi}[\overline{v}^*\overline{v}^* + \overline{v'v'}] \\
& - \frac{\tan\phi}{r}[\overline{v}][\overline{u}^*\overline{u}^* + \overline{u'u'}] + \frac{\partial[\overline{u}]}{\partial p}[\overline{\omega}^*\overline{u}^* + \overline{\omega'u'}] + \frac{\partial[\overline{v}]}{\partial p}[\overline{\omega}^*\overline{v}^* + \overline{\omega'v'}] + \Bigg\} + \\
& + \frac{1}{g}\Bigg\{ \overline{u'u'}^* \frac{1}{r\cos\phi}\frac{\partial\overline{u}^*}{\partial\lambda} + \overline{u'v'}^* \left( \frac{\partial\overline{u}^*}{r\partial\phi} + \overline{u}^*\frac{\tan\phi}{r} + \frac{1}{r\cos\phi}\frac{\partial\overline{v}^*}{\partial\lambda} \right) + \\
& + \overline{v'v'}^* \frac{\partial\overline{v}^*}{\partial\phi} - [\overline{u'u'}]^*\overline{v}^*\frac{\tan\phi}{r} \Bigg\}
\end{aligned}
$$

5  The curly brackets in $C_A$ and $C_K$ emphasize that the diagnostic module is able to distinguish between two components, one dealing with the conversion from/to zonal mean flow to/from eddy flow (first bracket), the other one dealing with the conversion among eddies (second bracket).

The source and sink terms, i.e. the generation/dissipation of APE and KE, are computed as residuals of the conversion terms at each reservoir.

10  *Competing interests.* The authors declare that they have no conflict of interest.

*Author contributions.* Valerio Lembo implemented the new direct method for the material entropy production, wrote the code and the paper draft. Frank Lunkeit implemented the first version of the LEC computation and supervised the whole code. Valerio Lucarini designed the diagnostic tool with its module partitioning and substantially contributed to the manuscript, editing the Introduction and Conclusions section.

*Acknowledgements.* We acknowledge the World Climate Research Programme's Working Group on Coupled Modelling, which is responsi-
15  ble for CMIP. For CMIP the U.S. Department of Energy's Program for Climate Model Diagnosis and Intercomparison provides coordinating support and led the development of software infrastructure in partnership with the Global Organization for Earth System Science Portals. Valerio Lembo was supported by the Collaborative Research Centre TRR181 "Energy Transfers in Atmosphere and Ocean" funded by the Deutsche Forschungsgemeinschaft (DFG, German Research Foundation), project No. 274762653. Valerio Lucarini was partially supported by the SFB/Transregio TRR181 project and by the EU Horizon2020 Blue Action Project.



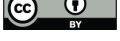

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





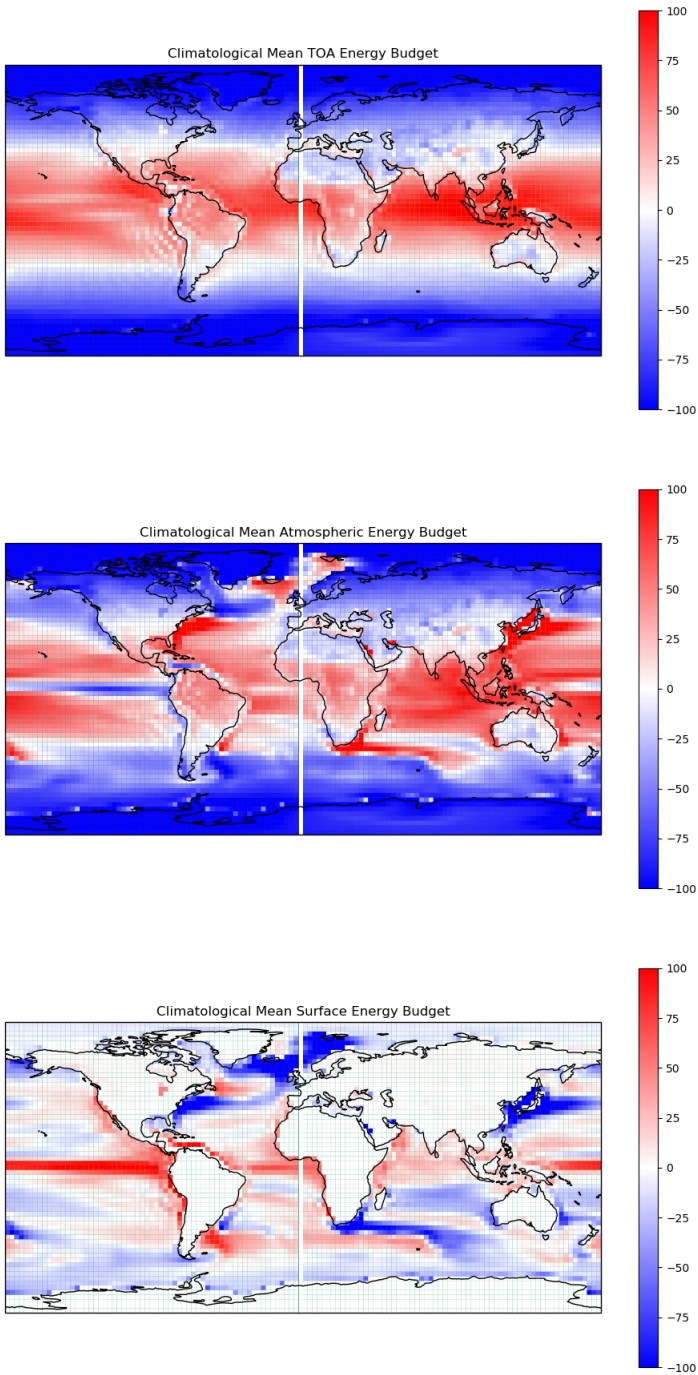

**Figure 1.** Climatological annual mean maps of (a) TOA, (b) atmospheric and (c) surface energy budgets for CanESM2 model (in $W/m^2$). The fluxes are positive when entering the domain, negative when exiting the domain.





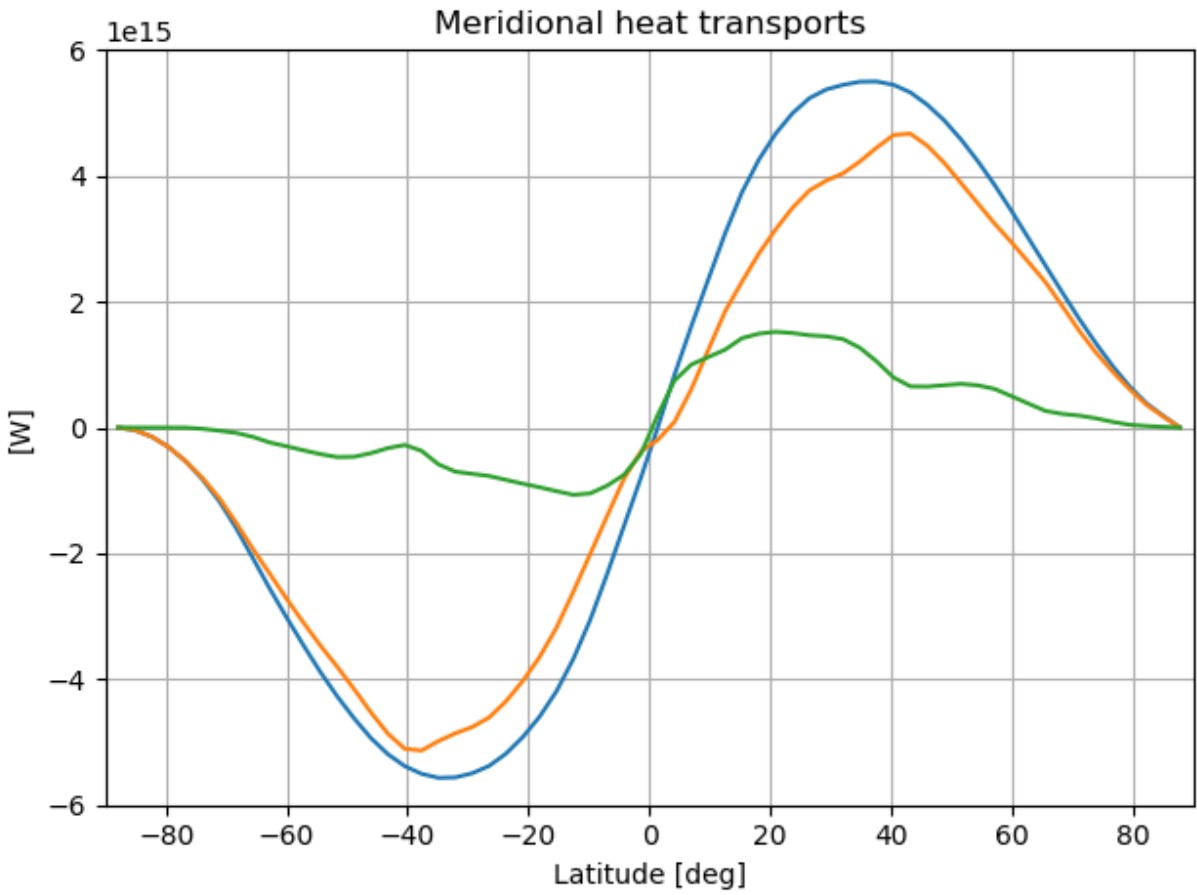

**Figure 2.** Climatological annual mean total (blue), atmospheric (orange) and oceanic (green) northward meridional enthalpy transports for 20 years of a pre-industrial CanESM2 model run (in $W$).



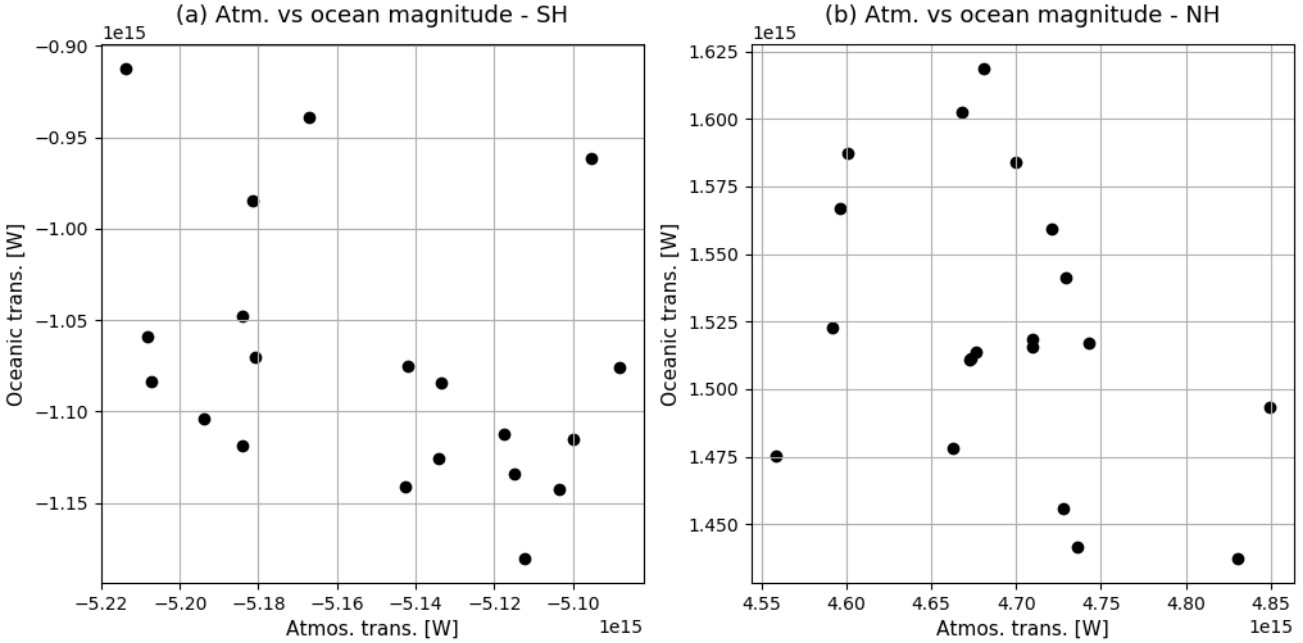

**Figure 3.** Scatter plots of 20 years pre-industrial CanESM2 annual mean atmospheric vs. oceanic peak magnitudes in the SH (a) and the NH (b) in $W$.

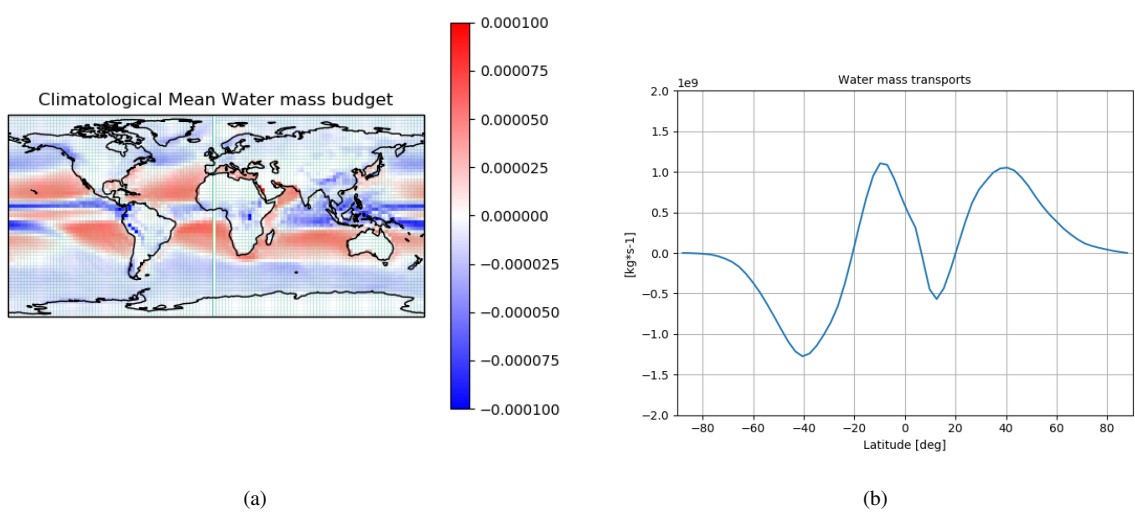

**Figure 4.** (a) Climatological annual mean water mass fluxes (in $kg * m^{-2} * s^{-1}$) and (b) annual mean northward meridional water mass transport (in in $kg * s^{-1}$) for a 20 years pre-industrial CanESM2 model run.





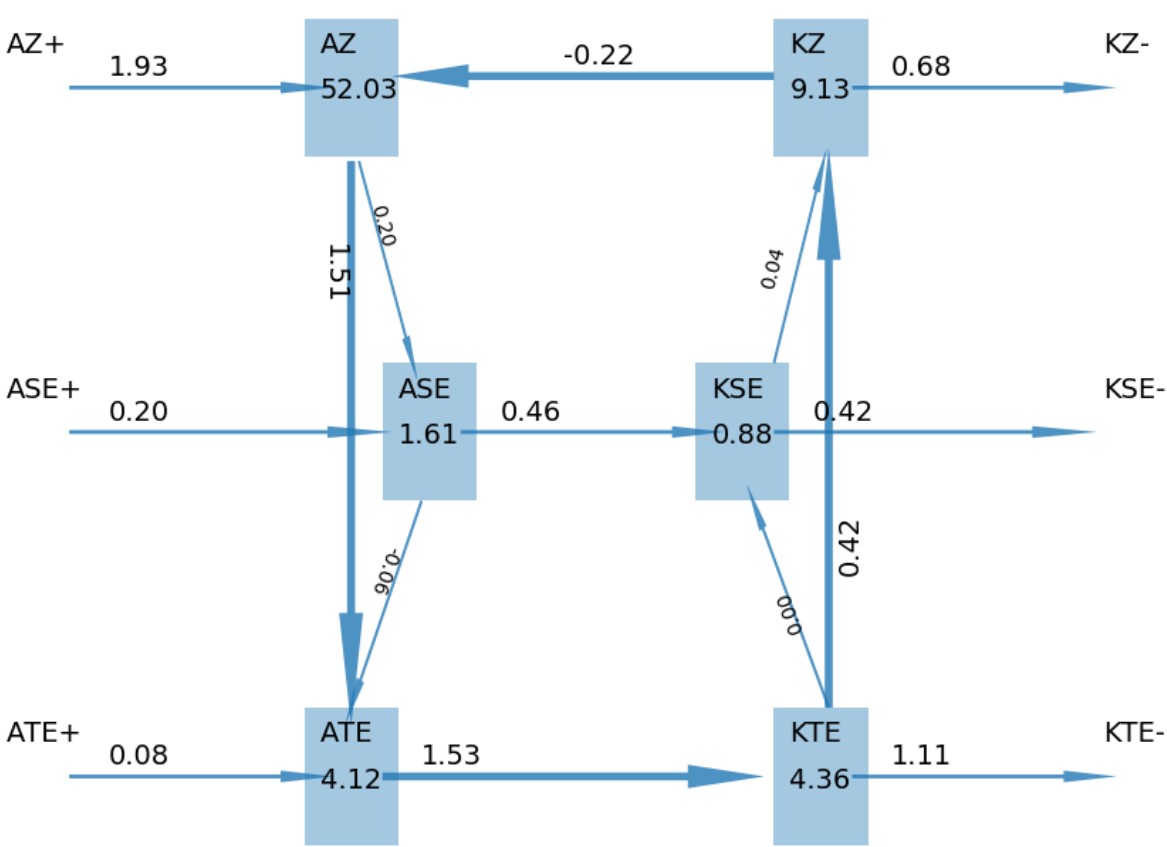

**Figure 5.** Diagram of Lorenz Energy Cycle (LEC) annual mean production, dissipation, storage and conversion terms for one year of pre-industrial CanESM2 model run. Reservoirs are displayed in units of $10^5 \; J * m^{-2}$, conversion terms as $W * m^{-2}$





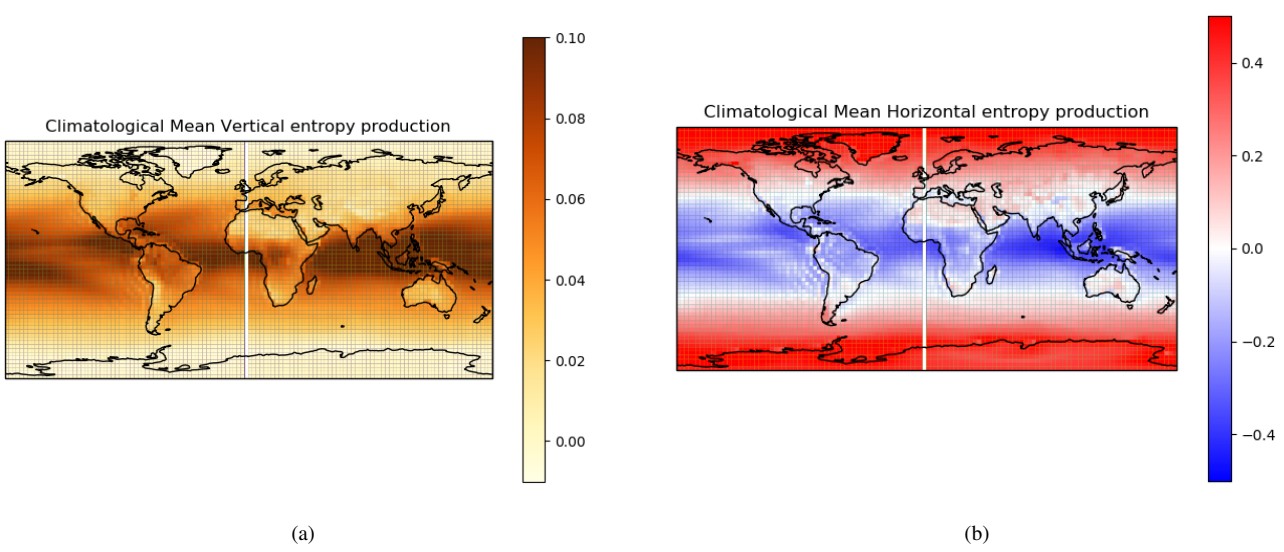

(a)                                              (b)

**Figure 6.** Climatological annual mean maps of (a) vertical component of the material entropy production (in $W/m^2 * K$), (b) horizontal component of the material entropy production (in $W/m^2 * K$), for a 20 years pre-industrial CanESM2 model run.





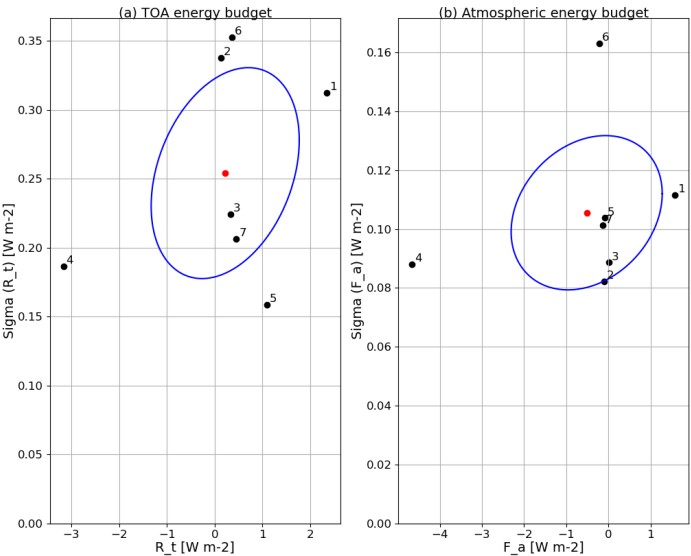

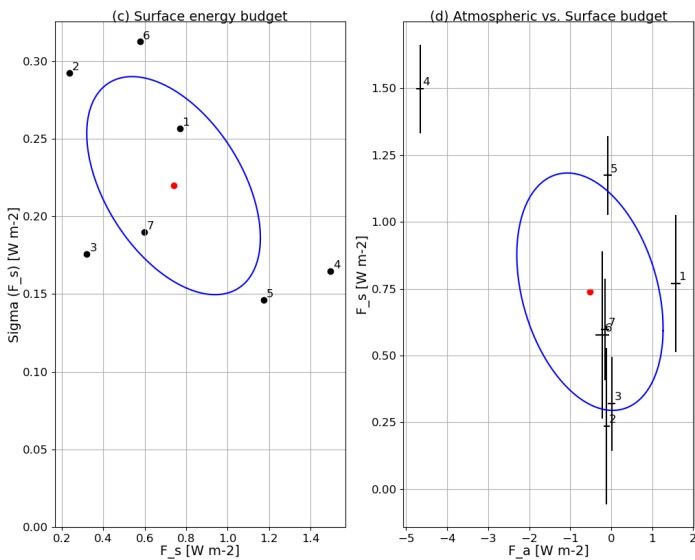

**Figure 7.** Multi-model ensemble scatter plots of annual mean averaged quantities vs. inter-annual variability in the piC scenario for: (a) TOA energy budget, (b) Atmospheric energy budget, (c) Surface energy budget. Panel d shows the atmospheric energy budget vs. the surface energy budget, with whiskers denoting the inter-annual variability as in panels (b) and (c). Blue ellipses denote the $\sigma$ standard deviation of the multi-model mean (denoted by the red dot). Model IDs are: 1. BNU, 2. Can2, 3. IPSL-M, 4. MIR5, 5. MIR-C, 6. MPI-LR, 7. MPI-MR.





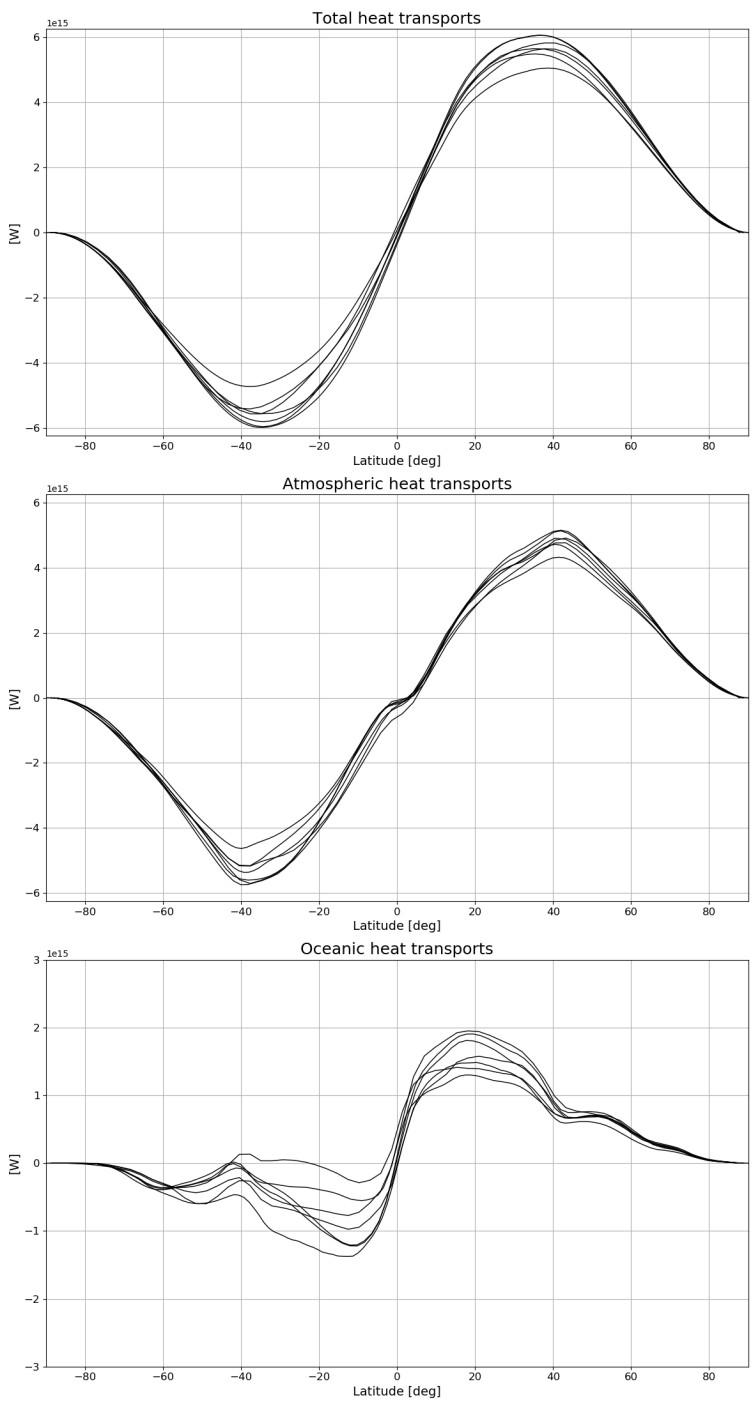

**Figure 8.** Climatological annual mean (a) total, (b) atmospheric, (c) oceanic northward meridional heat transports for all models (in $W$).





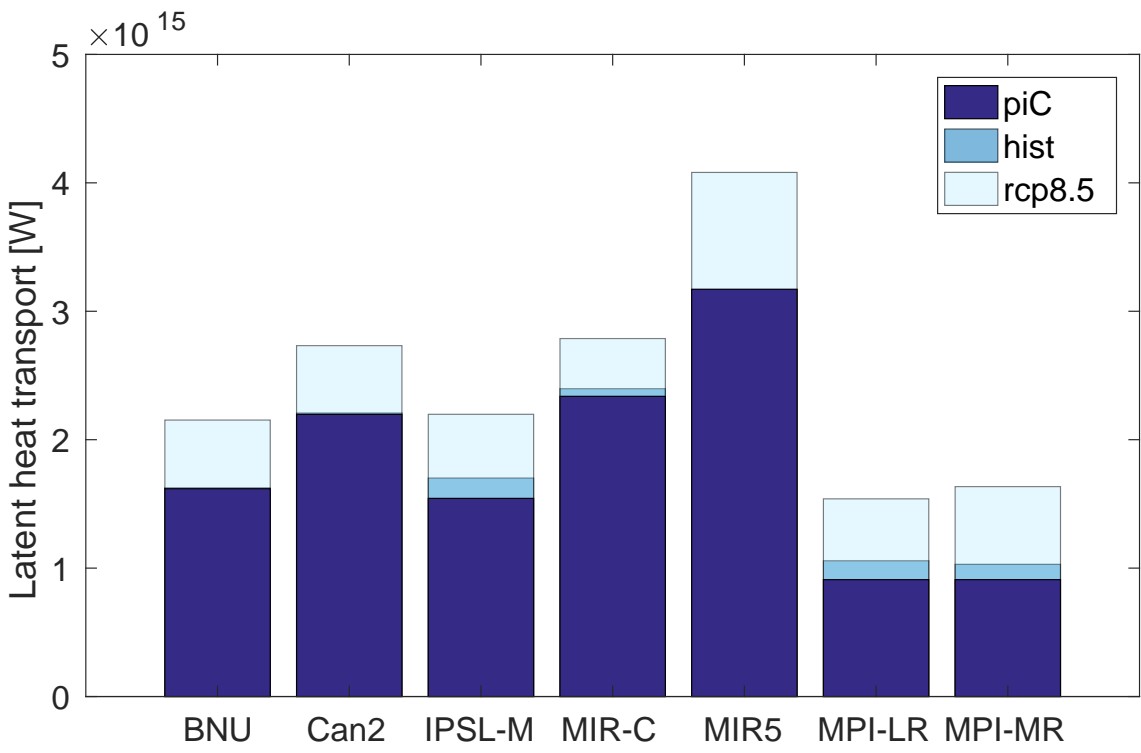

**Figure 9.** Latent heat transport between the Equator and 10N for the 7 models in the three scenarios. The transport increases from piC to RCP8.5. Values are in $W$ and are positive if northward directed.





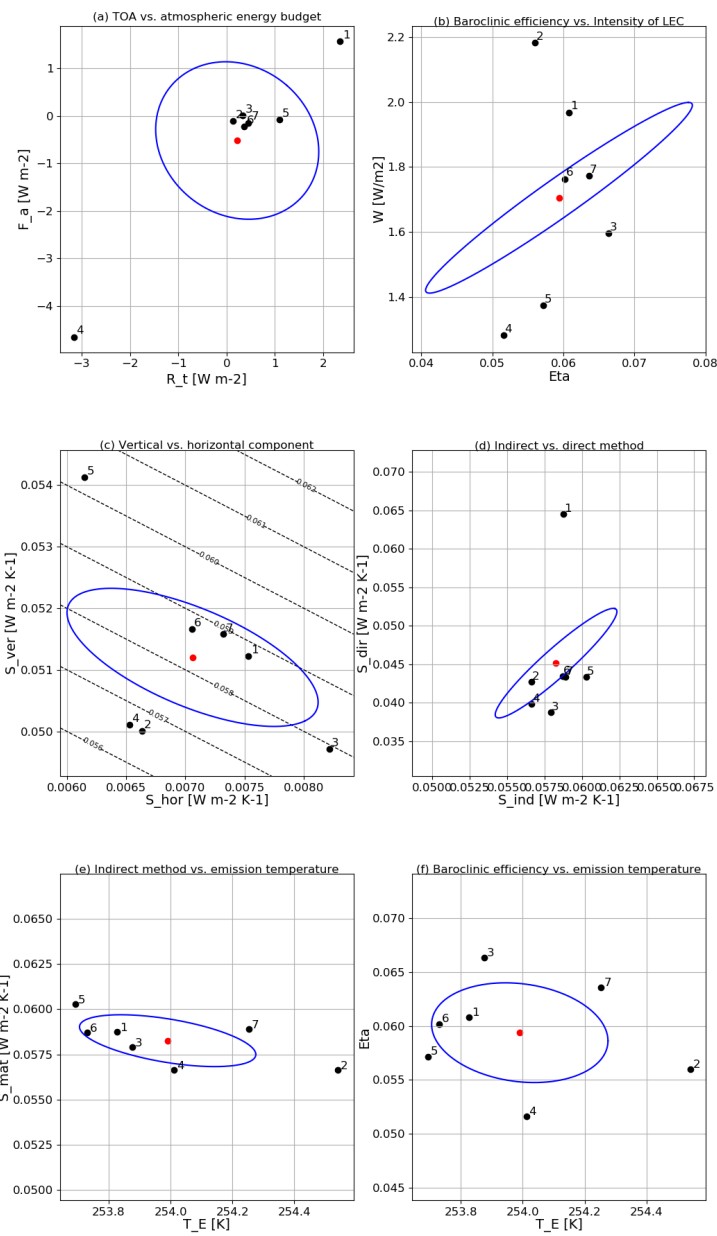

**Figure 10.** Multi-model ensemble scatter plots from piC scenario for: (a) Atmospheric energy budget vs. TOA energy budget (both in $W/m^2$), (b) Baroclinic efficiency vs. LEC intensity (in $W/m^2$), (c) vertical component of the material entropy production vs. vertical component (together with iso-lines of total material entropy production with the indirect method) (in $W/m^2*K$), (d) Direct vs. indirect method for material entropy production (in $W/m^2*K$). Blue ellipses denote the $\sigma$ standard deviation of the multi-model mean (denoted by the red dot). Model IDs are: 1. BNU, 2. Can2, 3. IPSL-M, 4. MIR5, 5. MIR-C, 6. MPI-LR, 7. MPI-MR.



**Table 1.** Variables used in the modules of the diagnostic tool

| Name (CMOR) | Time res. (min.) | Vert. res. | Energy budget | Hydrological cycle | LEC | MEP (indirect) | MEP (direct) |
|---|---|---|---|---|---|---|---|
| rsdt | monthly | 2D | X | | | X | |
| rsut | monthly | 2D | X | | | X | |
| rlut | monthly | 2D | X | | | X | |
| rlds | monthly | 2D | X | | | X | |
| rsds | monthly | 2D | X | | | X | |
| rlus | monthly | 2D | X | | | X | |
| rsus | monthly | 2D | X | | | X | |
| hfls | monthly | 2D | X | X | | X | X |
| hfss | monthly | 2D | X | | | X | X |
| pr[1] | monthly | 2D | | X | | | X |
| prsn[1] | monthly | 2D | | X | | | X |
| ta | day | 3D | | | X | | X |
| ua | day | 3D | | | X | | X |
| va | day | 3D | | | X | | X |
| wap | day | 3D | | | X | | X |
| tas | day | 2D | | | X | X | X |
| uas | day | 2D | | | X | X | X |
| vas | day | 2D | | | X | X | X |
| hus[3] | monthly | 3D | | | | | X |
| ts | monthly | 2D | | | | | X |
| ps | monthly | 2D | | | | | X |

[1]: precipitation fluxes are provided in units of $kg/m^2 s$. The program also accepts other units of measure and related fields, depending on the known formats by ESMValTool preprocessor.

[3]: specific humidity can also be given as near-surface 2-dimensional field (when available), or the lowermost pressure level of the 3-dimensional specific humidity field (variable name: huss).





**Table 2.** Annual mean values of a 20-years subset of control runs from 12 models participating to the CMIP5 Project for: TOA and surface energy budgets ($B_t$ and $B_s$, respectively), maximal and minimal peaks of atmospheric and oceanic meridional heat transports (with peak locations in latitude degrees specified in brackets) ($T_a^{max}$, $T_a^{min}$, $T_o^{max}$, $T_o^{min}$), water mass budget ($\overline{E-P}$), latent energy budget ($\overline{R_L}$), mechanical work by the Lorenz Energy Cycle, material entropy production computed with the direct and indirect methods ($\overline{\Sigma_{dir}^{mat}}$ and $\overline{\Sigma_{ind}^{mat}}$, respectively).

| | $B_t$ $\left(\frac{W}{m^2}\right)$ | $B_s$ $\left(\frac{W}{m^2}\right)$ | $T_a^{max}$ $(PW)$ | $T_a^{min}$ $(PW)$ | $T_o^{max}$ $(PW)$ | $T_o^{min}$ $(PW)$ | $\overline{E-P}$ $\left(\frac{kg}{m^2 s}\times 10^{-8}\right)$ | $\overline{R_L}$ $\left(\frac{W}{m^2}\right)$ | $W$ $\left(\frac{W}{m^2}\right)$ | $\overline{\Sigma_{dir}^{mat}}$ $\left(\frac{mW}{m^2 K}\right)$ | $\overline{\Sigma_{ind}^{mat}}$ $\left(\frac{mW}{m^2 K}\right)$ |
|---|---|---|---|---|---|---|---|---|---|---|---|
| BNU | 2.37 | 0.79 | 4.9 (42) | -5.1 (-39) | 1.9 (19) | -0.9(-17) | -207.1 | -5.89 | 2.0 | 64.9 | 58.7 |
| Can2 | 0.08 | 0.19 | 4.7 (41) | -5.1 (-39) | 1.5 (20) | -1.1 (-13) | 5.32 | -0.55 | 2.2 | 42.7 | 56.6 |
| IPSL-M | 0.33 | 0.32 | 4.6 (40) | -5.2 (-39) | 1.5 (19) | -1.4 (-14) | 11.1 | -0.48 | 1.6 | 38.7 | 57.9 |
| MIR-C | -3.16 | 1.50 | 4.8 (42) | -5.7 (-37) | 1.4 (19) | -0.4 (-9) | -1.24 | -0.70 | 1.3 | 39.8 | 56.5 |
| MIR5 | 1.06 | 1.13 | 4.2 (42) | -4.6 (-40) | 1.3 (18) | -0.6 (-10) | -2.94 | -0.71 | 1.4 | 43.4 | 60.3 |
| MPI-LR | 0.36 | 0.58 | 5.0 (42) | -5.5 (-38) | 1.9 (19) | -1.3 (-12) | -4.58 | -0.88 | 1.8 | 43.4 | 58.7 |
| MPI-MR | 0.45 | 0.60 | 5.1 (42) | -5.6 (-39) | 1.8 (19) | -1.3 (-11) | -4.03 | -0.86 | 1.7 | 43.4 | 58.9 |

**Table 3.** Same as in Table 2 for the period 1970-2000 of the historical runs.

| | $B_t$ $\left(\frac{W}{m^2}\right)$ | $B_s$ $\left(\frac{W}{m^2}\right)$ | $T_a^{max}$ $(PW)$ | $T_a^{min}$ $(PW)$ | $T_o^{max}$ $(PW)$ | $T_o^{min}$ $(PW)$ | $\overline{E-P}$ $\left(\frac{kg}{m^2 s}\times 10^{-8}\right)$ | $\overline{R_L}$ $\left(\frac{W}{m^2}\right)$ | $W$ $\left(\frac{W}{m^2}\right)$ | $\overline{\Sigma_{dir}^{mat}}$ $\left(\frac{mW}{m^2 K}\right)$ | $\overline{\Sigma_{ind}^{mat}}$ $\left(\frac{mW}{m^2 K}\right)$ |
|---|---|---|---|---|---|---|---|---|---|---|---|
| BNU | 2.94 | 1.41 | 4.9 (42) | -5.1 (-39) | 2.0 (19) | -0.8 (-13) | -199.9 | -5.66 | 1.9 | 63.9 | 60.0 |
| Can2 | 0.50 | 0.64 | 4.8 (42) | -5.2 (-39) | 1.6 (21) | -1.0 (-12) | 5.46 | -0.53 | 2.2 | 43.2 | 57.7 |
| IPSL-M | 0.92 | 0.89 | 4.7 (40) | -5.4 (-39) | 1.5 (19) | -1.4 (-13) | 10.4 | -0.47 | 1.6 | 39.6 | 59.1 |
| MIR-C | -2.71 | 1.90 | 4.9 (42) | -5.7 (-37) | 1.4 (17) | -0.8 (-11) | -1.12 | -0.69 | 1.2 | 40.0 | 57.4 |
| MIR5 | 1.30 | 1.32 | 4.3 (42) | -4.6 (-40) | 1.3 (18) | -0.6 (-9) | -1.38 | -0.67 | 1.3 | 43.4 | 61.1 |
| MPI-LR | 0.95 | 1.18 | 5.1 (42) | -5.6 (-38) | 1.9 (19) | -1.2 (-11) | -4.91 | -0.86 | 1.8 | 43.7 | 59.3 |
| MPI-MR | 0.99 | 1.11 | 5.1 (42) | -5.7 (-39) | 1.8 (19) | -1.2 (-11) | -3.91 | -0.82 | 1.7 | 43.7 | 59.5 |

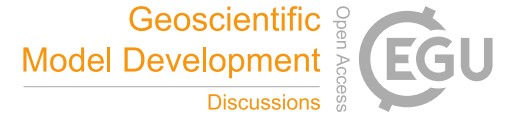

**Table 4.** Same as in Table 2 for the period 2071-2100 of the RCP8.5 runs.

| | $B_t$ $\left(\frac{W}{m^2}\right)$ | $B_s$ $\left(\frac{W}{m^2}\right)$ | $T_a^{max}$ $(PW)$ | $T_a^{min}$ $(PW)$ | $T_o^{max}$ $(PW)$ | $T_o^{min}$ $(PW)$ | $\overline{E-P}$ $\left(\frac{kg}{m^2 s} \times 10^{-8}\right)$ | $\overline{R_L}$ $\left(\frac{W}{m^2}\right)$ | $W$ $\left(\frac{W}{m^2}\right)$ | $\overline{\Sigma_{dir}^{mat}}$ $\left(\frac{mW}{m^2 K}\right)$ | $\overline{\Sigma_{ind}^{mat}}$ $\left(\frac{mW}{m^2 K}\right)$ |
|---|---|---|---|---|---|---|---|---|---|---|---|
| BNU | 4.79 | 3.19 | 5.5 (44) | -5.1 (-40) | 1.9 (17) | -0.8 (-11) | -146.1 | -4.15 | 1.8 | 61.7 | 65.5 |
| Can2 | 2.36 | 2.29 | 5.6 (42) | -5.4 (-39) | 1.4 (21) | -0.9 (-11) | 6.74 | -0.34 | 2.1 | 45.6 | 62.6 |
| IPSL-M | 2.79 | 2.63 | 4.9 (41) | -5.7 (-40) | 1.3 (17) | -1.3 (-12) | 7.38 | -0.38 | 1.7 | 43.4 | 65.3 |
| MIR-C | -0.29 | 4.11 | 4.9 (42) | -6.1 (-37) | 1.1 (15) | -0.9 (-8) | -1.13 | -0.51 | 1.1 | 42.5 | 63.1 |
| MIR5 | 3.28 | 3.11 | 4.4 (42) | -5.0 (-40) | 1.0 (18) | -0.7 (-7) | -0.94 | -0.58 | 1.2 | 45.0 | 65.8 |
| MPI-LR | 2.68 | 2.96 | 5.3 (42) | -6.0 (-37) | 1.8 (19) | -1.3 (-11) | -5.87 | -0.76 | 1.6 | 46.1 | 64.1 |
| MPI-MR | 2.61 | 2.72 | 5.4 (44) | -6.1 (-40) | 1.7 (17) | -1.3 (-11) | -3.78 | -0.68 | 1.7 | 46.0 | 64.3 |

**Table 5.** Annual mean land-ocean asymmetries for atmospheric, latent energy budget and the difference of the two, for 20 years of multi-model ensemble simulations under piC conditions. Values are in $W/m^2$.

| | $B\_a$ $[W/m^2]$ | | $R\_L$ $[W/m^2]$ | | $(B\_a - R\_L)$ $[W/m^2]$ | |
|---|---|---|---|---|---|---|
| | Oc. | Land | Oc. | Land | Oc. | Land |
| BNU | 5.6 | -10.7 | 5.8 | -25 | -0.2 | 14.3 |
| Can2 | 6.3 | -21.2 | 7.4 | -18 | -1.1 | 3.2 |
| IPSL-M | 4.0 | -11 | 7.3 | -15.1 | -3.3 | 4.1 |
| MIR-C | 0.2 | -16.8 | 7.8 | -21.8 | -7.6 | 5.0 |
| MIR5 | 3.8 | -11 | 11.8 | -25.7 | -8.0 | 14.7 |
| MPI-LR | 5.7 | -20.5 | 6.1 | -17.9 | -0.4 | 2.6 |
| MPI-MR | 5.6 | -20.2 | 6.2 | -18.5 | -0.6 | 1.7 |





**Table 6.** Evolution of land-ocean asymmetries for latent energy and for the residual of the atmospheric budget, for 20 years of multi-model ensemble simulations under the two extremal scenarios (piC and RCP8.5). Values are in $PW$, and are positive if they are directed toward land. Values in brackets are from spatial integration over land, those not in brackets from integration over oceans.

|  | $R\_L$ [PW] |  | $(B\_a - R\_L)$ [PW] |  |
| --- | --- | --- | --- | --- |
|  | piC | rcp8.5 | piC | rcp8.5 |
| BNU | 2.09 (3.72) | 2.91 (3.59) | -0.07 ( 2.13) | -1.34 (2.29) |
| Can2 | 2.67 (2.81) | 3.32 (3.34) | -0.39 (-0.47) | -1.17 (0.37) |
| IPSL-M | 2.63 (2.24) | 3.07 (2.54) | -1.19 ( 0.64) | -1.75 (0.91) |
| MIR-C | 2.82 (3.26) | 3.21 (3.58) | -2.76 ( 0.76) | -3.19 (1.51) |
| MIR5 | 4.27 (3.83) | 4.48 (3.94) | -2.90 ( 2.19) | -3.58 (2.78) |
| MPI-LR | 2.18 (2.67) | 2.63 (3.07) | -0.11 (-0.39) | -0.23 (0.16) |
| MPI-MR | 2.32 (2.76) | 2.81 (3.22) | -0.22 (-0.25) | -0.80 (0.26) |





**Table 7.** Annual mean values of APE and KE reservoirs and conversion terms in the LEC. Values are in $10^5$ $J/m^2$ for the reservoirs , in $W/m^2$ for the conversion terms. For the notation, refer to Appendix A1

|  | $A_z$ | $A_e$ | $K_z$ | $K_e$ | $C(A\_z, K\_z)$ | $C(A\_e, K\_e)$ | $C(A\_z, A\_e)$ | $C(K\_e, K\_z)$ |
|---|---|---|---|---|---|---|---|---|
| piC |  |  |  |  |  |  |  |  |
| BNU | 54.9 | 5.8 | 9.2 | 5.6 | -0.20 | 1.7 | 1.5 | 0.50 |
| Can2 | 52.2 | 5.7 | 8.8 | 5.2 | -0.20 | 1.9 | 1.7 | 0.40 |
| IPSL-M | 52.0 | 5.7 | 8.8 | 5.2 | -0.20 | 1.6 | 1.4 | 0.40 |
| MIR-C | 53.1 | 5.7 | 7.6 | 5.3 | -0.03 | 1.3 | 1.2 | 0.20 |
| MIR5 | 48.0 | 5.9 | 5.9 | 5.0 | -0.05 | 1.3 | 1.2 | 0.20 |
| MPI-LR | 46.8 | 5.6 | 7.2 | 5.8 | -0.10 | 1.6 | 1.5 | 0.40 |
| MPI-MR | 48.8 | 5.6 | 7.1 | 6.1 | -0.10 | 1.6 | 1.6 | 0.40 |
| hist |  |  |  |  |  |  |  |  |
| BNU | 52,8 | 5.7 | 10.4 | 5.8 | -0.20 | 1.7 | 1.5 | 0.5 |
| Can2 | 50.7 | 5.6 | 10.0 | 5.4 | -0.20 | 1.9 | 1.6 | 0.5 |
| IPSL-M | 50.5 | 5.9 | 10.5 | 5.4 | -0.20 | 1.9 | 1.6 | 0.50 |
| MIR-C | 51.4 | 5.8 | 8.1 | 5.3 | -0.02 | 1.2 | 1.6 | 0.40 |
| MIR5 | 47.2 | 5.8 | 6.4 | 5.1 | -0.05 | 1.3 | 1.2 | 0.30 |
| MPI-LR | 45.6 | 5.5 | 7.6 | 6.0 | -0.15 | 1.6 | 1.5 | 0.50 |
| MPI-MR | 46.9 | 5.6 | 7.6 | 6.3 | -0.14 | 1.6 | 1.5 | 0.50 |
| rcp8.5 |  |  |  |  |  |  |  |  |
| BNU | 46.9 | 5.3 | 12.5 | 6.1 | -0.20 | 1.6 | 1.3 | 0.50 |
| Can2 | 49.1 | 5.4 | 13.1 | 5.9 | -0.20 | 1.9 | 1.5 | 0.4 |
| IPSL-M | 47.4 | 5.6 | 14.8 | 6.3 | 0.05 | 1.8 | 1.4 | 0.50 |
| MIR-C | 50.0 | 5.4 | 11.5 | 5.6 | 0.01 | 1.1 | 1.3 | 0.40 |
| MIR5 | 47.2 | 5.5 | 9.1 | 5.4 | -0.03 | 1.2 | 1.0 | 0.3 |
| MPI-LR | 44.0 | 5.3 | 10.4 | 6.7 | -0.10 | 1.6 | 1.4 | 0.60 |
| MPI-MR | 44.8 | 5.4 | 9.8 | 6.8 | -0.10 | 1.6 | 1.4 | 0.60 |

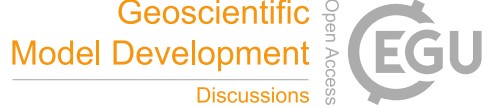



**Table 8.** Annual mean components of the material entropy production, obtained with the direct and indirect methods. Values are in $mW * m^{-2} * K^{-1}$.

|        | $S_{hyd}$ | $S_{sens}$ | $S_{kin}$ | $S_{ver}$ | $S_{hor}$ |
|--------|-----------|------------|-----------|-----------|-----------|
| BNU    | 54.8      | 2.67       | 6.97      | 51.2      | 7.5       |
|        | 54.6      | 2.59       | 6.79      | 53.3      | 7.4       |
|        | 52.9      | 2.40       | 6.38      | 58.7      | 6.8       |
| Can2   | 32.1      | 2.97       | 7.70      | 50.0      | 7.5       |
|        | 32.8      | 2.85       | 7.58      | 51.5      | 6.6       |
|        | 35.9      | 2.64       | 7.18      | 56.7      | 5.96      |
| IPSL-M | 30.1      | 2.92       | 5.6       | 49.7      | 8.2       |
|        | 31.0      | 2.84       | 5.71      | 51.0      | 8.3       |
|        | 34.8      | 2.64       | 6.01      | 57.1      | 8.2       |
| MIR-C  | 32.7      | 2.53       | 4.53      | 50.1      | 6.53      |
|        | 33.0      | 2.47       | 4.44      | 50.9      | 6.52      |
|        | 36.2      | 2.30       | 3.96      | 57.0      | 6.11      |
| MIR5   | 36.7      | 1.84       | 4.87      | 54.1      | 6.19      |
|        | 36.9      | 1.80       | 4.75      | 55.0      | 6.18      |
|        | 39.1      | 1.71       | 4.24      | 59.8      | 6.0       |
| MPI-LR | 34.7      | 2.51       | 6.23      | 51.6      | 7.05      |
|        | 35.2      | 2.41       | 6.12      | 52.6      | 7.00      |
|        | 38.2      | 2.23       | 5.66      | 57.3      | 6.82      |
| MPI-MR | 34.6      | 2.52       | 6.26      | 51.6      | 7.32      |
|        | 35.1      | 2.41       | 6.14      | 52.4      | 7.27      |
|        | 37.9      | 2.23       | 5.84      | 57.2      | 7.15      |





**Table 9.** 20-year annual mean material entropy production associated with the hydrological cycle. Each component denotes a different process: (from left to right) evaporation, rainfall precipitation, snowfall precipitation, snow melting at the ground, potential energy of the droplet. Values are in $mW * m^{-2} * K^{-1}$.

|  | $S_e$ | $S_r$ | $S_s$ | $S_m$ | $S_p$ |
|---|---|---|---|---|---|
| | -278.7 | 307.5 | 24.1 | -2.58 | 4.53 |
| BNU | -280.7 | 310.1 | 23.2 | -2.47 | 4.65 |
| | -296.9 | 329.3 | 17.2 | -1.83 | 5.16 |
| | -269.2 | 276.7 | 25.9 | -2.49 | 4.21 |
| Can2 | -269.5 | 277.6 | 22.6 | -2.42 | 4.29 |
| | -282.5 | 298.0 | 17.4 | -1.86 | 4.79 |
| | -271.5 | 274.5 | 25.9 | -2.78 | 4.02 |
| IPSL-M | -274.8 | 279.5 | 24.8 | -2.66 | 4.13 |
| | -293.3 | 306.2 | 19.4 | -2.08 | 4.65 |
| | -278.6 | 283.1 | 22.8 | 2.46 | 4.17 |
| MIR-C | -271.5 | 280.3 | 22.4 | 2.41 | 4.21 |
| | -286.7 | 303.5 | 16.5 | -1.77 | 4.70 |
| | -315.6 | 328.0 | 21.8 | -2.35 | 4.86 |
| MIR5 | -313.7 | 326.6 | 21.4 | -2.31 | 4.91 |
| | -321.7 | 338.7 | 18.8 | -2.03 | 4.65 |
| | -288.1 | 295.2 | 26.0 | -2.80 | 4,46 |
| MPI-LR | -287.6 | 296.1 | 24.9 | -2.68 | 4.53 |
| | -298.2 | 313.0 | 20.7 | -2.22 | 4.97 |
| | -291.9 | 299.1 | 25.7 | -2.76 | 4.43 |
| MPI-MR | -292.2 | 300.8 | 24.6 | -2.65 | 4.52 |
| | -303.2 | 318.3 | 20.1 | -2.17 | 4.94 |



**Table 10.** 20-year Annual mean irreversibility and baroclinic efficiency for each model and each scenario. Baroclinic efficiency is rescaled as $10^{-2}$.

|  |  | piC | hist | rcp8.5 |
|---|---|---|---|---|
| BNU | $\alpha$ | 8.6 | 8.8 | 9.0 |
|  | $\eta$ | 6.1 | 6.1 | 5.6 |
| Can2 | $\alpha$ | 4.8 | 4.9 | 5.6 |
|  | $\eta$ | 5.6 | 5.7 | 5.1 |
| IPSL-M | $\alpha$ | 6.4 | 6.4 | 6.6 |
|  | $\eta$ | 6.6 | 6.7 | 6.4 |
| MIR-C | $\alpha$ | 8.4 | 8.5 | 10.1 |
|  | $\eta$ | 5.1 | 5.1 | 4.8 |
| MIR5 | $\alpha$ | 8.4 | 8.6 | 10.1 |
|  | $\eta$ | 5.7 | 5.8 | 5.6 |
| MPI-LR | $\alpha$ | 6.4 | 6.3 | 7.5 |
|  | $\eta$ | 6.0 | 6.1 | 5.8 |
| MPI-MR | $\alpha$ | 6.4 | 6.2 | 7.2 |
|  | $\eta$ | 6.3 | 6.4 | 6.2 |