# Peer review of "TheDiaTo (v1.0) – A new diagnostic tool for water, energy and entropy budgets in climate models"

_Geoscientific Model Development, 2019_

## Referee Comment (RC1) · Anonymous Referee #1 · 12 May 2019

The study 'TheDiaTo (v1.0) – A new diagnostic tool for water, energy and entropy budgets in climate models'Âăby Lembo et al describes Thermodynamic Diagnostic Tool (TheDiaTo), a new diagnostic tool that evaluates the energy budget, the hydrological cycle, the Lorenz Energy Cycle and the material entropy production. This is a very useful tool for evaluating climate models at global scale, as well as over land and ocean. The strength of this tool is its flexibility, so it can be run over a large multi-model ensemble (such as CMIP) and give a robust overview of the performance of the models. The paper describes the components of the tool and their results well. The figures and tables are clear.

Specific comments:

- Introduction: For energy and water budget studies using observations, please add

L'Ecuyer et al, 2015 and Rodell et al, 2015. For studies evaluating the energy and water budgets in climate models, please add Demory et al, 2014; Terai et al, 2017; Vanniere et al, 2018 (e.g. in P2 L29, P3 L22, P4 paragraph 1.2). These studies highlighted the need for more robust evaluation of these budgets in climate models.

- section 2 data and software requirements: are calculations performed on the native grid of the models or on a common grid?

- section 3: can the tool consider observational data, where available, to validate the models?

- section 3: can the tool also work with seasonal means?

- section 3.2 hydrological cycle: Assessing the water budget in climate models is particularly important to determine their conservation error and its evolution throughout the simulation. A model may have a conservation error but that remains stable over time, while another one may constantly lose/gain mass over time (as noted in Liepert and Previdi, 2012). In this study, this would be particularly important to understand what is happening in BNU. Is it possible to see the time evolution of the conservation error with TheDiaTo (Fig. 2 of Liepert and Previdi), and if not, would it be possible to add it? This would be beneficial for a thorough evaluation of the climate models.

- P15 L21: is there a reason for picking this period: 2441-2460?

- P18 L28 & L32: 'bias' generally refers to errors relative to observations. Here, it may be more robust to refer to 'imbalance'.

- Tables 8 and 9: specify that rows are for PiC, hist and rcp85.

There are several typos throughout the paper. I noted a few here, but there are surely others: - P1 L5: replace 'Top-of-Atmosphere (TOA)' by TOA, as defined just earlier - P2 L6: replace by 'cryosphere' - P2 L16: remove 'of things' - P2 L21: be more specific with 'the required standards mentioned above' so we don't have to jump back to recall these - P2 L23: same as above for 'enunciated above ideas' - P5 L5: same as above

for 'mentioned aspects' - P7 L18 & P15 L26: replace 'EB' by energy budget - P10 L 13: 'lhs' -> left-hand side - P12 L10: 'rhs' -> right-hand side - P17 L13 'edddy' -> 'eddy' - P20 L26: correct to 'orographically' - P22 L25: 'rato' -> 'ratio' - P22 L30: 'use' -> 'used' - P23 L17: 'effected' -> 'affected' - P24 L25: 'loking' -> 'looking' - P25 L15: 'vertcial' -> 'vertical'

References: - Demory ME, Vidale PL, Roberts MJ, Berrisford P, Strachan J, Schiemann R, Mizielinski MS (2014) The role of horizontal resolution in simulating drivers of the global hydrological cycle. Climate Dynamics 42(7-8):2201–2225, DOI 10.1007/s00382-013-1924-4

- L'Ecuyer TS, Beaudoing HK, Rodell M, Olson W, Lin B, Kato S, Clayson CA, Wood E, Sheeld J, Adler R, Hu↵man G, Bosilovich M, Gu G, Robertson F, Houser PR, Chambers D, Famiglietti JS, Fetzer E, Liu WT, Gao X, Schlosser CA, Clark E, Lettenmaier DP, Hilburn K (2015) The observed state of the energy budget in the early twenty-first century. Journal of Climate 28(21):8319– 8346, DOI 10.1175/jcli-d-14-00556.1

- M. Rodell, H. K. Beaudoing, T. S. L'Ecuyer, W. S. Olson, J. S. Famiglietti, P. R. Houser, R. Adler, M. G. Bosilovich, C. A. Clayson, D. Chambers, E. Clark, E. J. Fetzer, X. Gao, G. Gu, K. Hilburn, G. J. Huffman, D. P. Lettenmaier, W. T. Liu, F. R. Robertson, C. A. Schlosser, J. Sheffield, and E. F. Wood. The Observed State of the Water Cycle in the Early Twenty-First Century. J. Clim., 28:8289–8318, 2015.

- Terai, C.R., Caldwell, P.M., Klein, S.A. et al. (2017). The atmospheric hydrologic cycle in the ACME v0.3 model. Clim Dyn. https://doi.org/10.1007/s00382-017-3803-x

- Vannière, B., Demory, ME., Vidale, P.L. et al. Clim Dyn (2018). https://doi.org/10.1007/s00382-018-4547-y

---

## Referee Comment (RC2) · Anonymous Referee #2 · 8 Jul 2019

This paper describes a suite of software tools that are intended to calculate a range of quite sophisticated diagnostics relating to the energy and entropy budgets of the global atmosphere and ocean circulations from large-scale numerical models, and to aspects of the hydrological cycle. The overall thrust of the computations are based on a thermodynamic interpretation of the climate system as a complex heat engine, allowing for energy and entropy fluxes in various forms at a level of sophistication that matches that of the most recent Earth System models. The authors have a strong track record of research in this field and have published widely on various aspects of these diagnostics as applied to a number of existing climate model simulations. The present tool would seem to offer the possibility of others carrying out similar analyses. This is to be commended in the interests of aiding model intercomparisons.

[Figure]

The range of options being offered seems fairly comprehensive, the only significant restrictions appearing to be the assumption of hydrostatic balance in the dynamical equations. The technical details are described fairly fully in a set of appendices, and some typical results are presented in the later sections through applications of the tool to CMIP5 model runs under three different scenarios, representing pre-industrial conditions, historic anthropogenic forcing and a future "business as usual" projection. This serves well to illustrate the potential insights provided by the diagnostics on offer.

The paper overall is well structured and reasonably well written, although the English in a few places is a little "clunky" with occasional words missing or with awkward phrasing, which may tend to distract some readers. My main criticism (speaking as an "outsider" - not directly involved in diagnostic studies of state of the art Earth climate models) would be of the description in Section 2 of the data requirements for use of the tool. This is quite short and seems to presume quite a lot of knowledge on the part of the reader (unless they are prepared to spend a lot of time and effort consulting the documentation for the ESMValTool software, referenced as Eyring et al. (2016b)). Table 1 lists the variables required to be presented to the tool for the various calculations, but simply names the variables according to the CMOR convention without explanation. This is not helpful if you don't know that convention, and it is not explained anywhere in the paper except in external references. Why not provide a key for these variables or an explanation in another column of Table 1? Given the general applicability of much of the tool to climates that are not restricted to Earth (as acknowledged in the paper itself), readers would appreciate a bit more help and guidance for novice users. Is there a graphical interface, for example, or is it run via a shell script? Some more guidance here and perhaps an example script would be helpful, either in the text or another appendix.

In other respects, this looks to be a potentially useful facility for anyone interested in model inter comparisons and/or complex climate diagnostics, and could be even more attractive if made a little more user-friendly. I outline below a few more minor points

[Figure]

and suggestions.

- Figure 5 and associated text: The reservoir terms ASE, ATE, KSE and KTE are not explained or defined very obviously. They presumably refer to stationary and transient components of AE and KE? It would be helpful to state this in the caption and perhaps mention in the text?

- P.17 line 23 as example of awkward phrasing: should probably read "...as a possible reason for the well known cold pole bias....". But there are quite a few others - too numerous to list in detail. Perhaps recruit someone whose first language is English to proofread - or the authors should proofread more thoroughly before submission!

- Presumably this tool could be used on reanalysis datasets too? Perhaps comment further?

---

## Author Comment (AC1) · 22 Jul 2019

**Reply to comments by Referee #1**

The study "TheDiaTo (v1.0) – A new diagnostic tool for water, energy and entropy budgets in climate models" by Lembo et al describes Thermodynamic Diagnostic Tool (TheDiaTo), a new diagnostic tool that evaluates the energy budget, the hydrological cycle, the Lorenz Energy Cycle and the material entropy production. This is a very useful tool for evaluating climate models at global scale, as well as over land and ocean. The strength of this tool is its flexibility, so it can be run over a large multi-model ensemble (such as CMIP) and give a robust overview of the performance of the models. The paper describes the components of the tool and their results well. The figures and tables are clear.

Specific comments:

- Introduction: For energy and water budget studies using observations, please add L'Ecuyer et al, 2015 and Rodell et al, 2015. For studies evaluating the energy and water budgets in climate models, please add Demory et al, 2014; Terai et al, 2017; Vanniere et al, 2018 (e.g. in P2 L29, P3 L22, P4 paragraph 1.2).  These studies highlighted the need for more robust evaluation of these budgets in climate models.

*Thank you for the suggestions. We have added these references in a new sentence in Sect. 1.2: "Water mass budget has been assessed in observations (L'Ecuyer et al., 2015; Rodell et al., 2015), as well as in climate models, focusing on the hydrological cycle alone (Terai et al., 2018) or evaluating it in conjunction with the energy budgets (Demory et al., 2014; Vannière et al., 2019)."*

- section 2 data and software requirements: are calculations performed on the native grid of the models or on a common grid?

*The calculations are performed on the native grid of the model, as provided in the CMIP repository. We specified this in Sect. 2 by adding the following sentence: "For model inter-comparisons (Section 5), computations are performed on the native grid of the model."*

- section 3: can the tool consider observational data, where available, to validate the models?

*The tool can be used to compare models and observational data, as long as the data are global-scale and on a regular grid. The ESMValTool v.2 framework allows for straightforward comparison with Reanalyses (as it will be discussed in the ESMValTool v.2 report that is soon to be submitted on this same journal). Since referee #1 also suggested to mention that, we have added a comment on the range of applicability to observational-based gridded datasets in Sect. 2 as follows: "Therefore, the tool is suitable for the evaluation of any kind of gridded datasets, provided that they contain the necessary variables on a regular grid, including blends of observations and Reanalyses. In our description of the software features, we focus on model evaluation and multi-model intercomparison."*

- section 3: can the tool also work with seasonal means?

*Technically, no constraint is given on the usage of a subset of months. Although this would cause in principle no problem on the computations of the budgets and the LEC strength, the material entropy production would be inconsistent, because there is no straightforward argument justifying that the indirect method would provide a reasonable estimate of this quantity (as mentioned when discussing Eq. 11 in Sect.  3.4). A thorough explanation of the assumptions is provided in Lucarini et al. 2011. Furthermore, the implied transports rely on the fact that the differential heating is balanced by the oceans and the atmosphere, but this is not certainly accurate*

*at the smaller than annual timescale (cfr. for instance Rose and Ferreira 2013 on the validity of the Bjerknes compensation mechanism).*

- section 3.2 hydrological cycle: Assessing the water budget in climate models is particularly important to determine their conservation error and its evolution throughout the simulation. A model may have a conservation error but that remains stable over time, while another one may constantly lose/gain mass over time (as noted in Liepert and Previdi, 2012). In this study, this would be particularly important to understand what is happening in BNU. Is it possible to see the time evolution of the conservation error with TheDiaTo (Fig. 2 of Liepert and Previdi), and if not, would it be possible to add it? This would be beneficial for a thorough evaluation of the climate models.

*The diagnostic tool allows to obtain annual mean time series of energy and water mass budgets for the Norther Hemisphere, the Southern Hemisphere and the global mean. For sake of brevity, we decided not to include them. An example of what the time series for the BNU-ESM model would look like is shown in Figures 1-3. It can be noticed that the conservation error is quite massive, compared to the other models, but relatively constant.*

[Figure]

**Figure 1:** *annual mean time series of the energy budget for BNU-ESM pre-industrial run. Red denotes Southern Hemisphere, blue denotes Northern Hemisphere, black the global mean.*

[Figure]

**Figure 2:** *same as Figure 1, for the last 20 years of the historical experiment.*

[Figure]

**Figure 3:** *same as Figure 1, for the last 20 years of the RCP8.5 experiment.*

- P15 L21: is there a reason for picking this period: 2441-2460?

*The reason for that is that there is no other section of the pre-industrial CanESM2 run for which 20-years of all the needed variables is available on the repository. We explained this by adding the following sentence: "The choice of the sub-period is motivated by the fact that it is the only part of the run for which a 20-years subsequent dataset of the needed variables is available in the repository."*

- P18 L28 & L32: 'bias' generally refers to errors relative to observations. Here, it may be more robust to refer to 'imbalance'.

*Thank you. We replaced "bias" with the more appropriate term "imbalance".*

- Tables 8 and 9: specify that rows are for PiC, hist and rcp85.

*We have changed the table captions as follows: "For each model, the first row denotes the estimates from piC, the second row from hist, the third row from rcp8.5".*

There are several typos throughout the paper. I noted a few here, but there are surely others:
- P1 L5: replace 'Top-of-Atmosphere (TOA)' by TOA, as defined just earlier

*Done. Thank you.*

- P2 L6: replace by 'cryosphere'

*Done. Thank you.*

- P2 L16: remove 'of things'

*Done. Thank you.*

- P2 L21: be more specific with 'the required standards mentioned above' so we don't have to jump back to recall these

*We have replaced "prepared according to the required standards mentioned above" with "containing process-based and end-user-relevant diagnostics".*

- P2 L23: same as above for 'enunciated above ideas'

*We have added "reaching a steady state as a result of an interplay of multi-scale processes".*

- P5 L5: same as above C2 for 'mentioned aspects'

*We have added "(i.e. the energy and water mass budgets and meridional transports, the LEC strength and the material entropy production)".*

- P7 L18 & P15 L26: replace 'EB' by energy budget

*Done. Thank you.*

- P10 L 13: 'lhs' -> left-hand side

*Done. Thank you.*

- P12 L10: 'rhs' -> right-hand side

*Done. Thank you.*

- P17 L13 'edddy' -> 'eddy'

*Done. Thank you.*

- P20 L26: correct to 'orographically'

*We believe that the correct spelling is "orografically".*

- P22 L25: 'rato' -> 'ratio'

*Done. Thank you.*

- P22 L30: 'use' -> 'used'

*Done. Thank you.*

- P23 L17: 'effected' -> 'affected'

*What we mean here is that baroclinic eddies "effect" baroclinic conversions. They can be both "synoptic-scale" and "planetary-scale" eddies, as shown in a recent paper of us (Lembo et al. 2019, 46, Geophys. Res. Lett.). We have added this reference, since it was not yet accepted by the time the manuscript was submitted.*

- P24 L25: 'loking' -> 'looking'

*Done. Thank you.*

- P25 L15: 'vertcial' -> 'vertical'

*Done. Thank you.*

---

## Author Comment (AC2) · 22 Jul 2019

**Reply to comments by Referee #2**

This paper describes a suite of software tools that are intended to calculate a range of quite sophisticated diagnostics relating to the energy and entropy budgets of the global atmosphere and ocean circulations from large-scale numerical models, and to aspects of the hydrological cycle. The overall thrust of the computations are based on a thermodynamic interpretation of the climate system as a complex heat engine, allowing for energy and entropy fluxes in various forms at a level of sophistication that matches that of the most recent Earth System models. The authors have a strong track record of research in this field and have published widely on various aspects of these diagnostics as applied to a number of existing climate model simulations. The present tool would seem to offer the possibility of others carrying out similar analyses. This is to be commended in the interests of aiding model intercomparisons.

The range of options being offered seems fairly comprehensive, the only significant restrictions appearing to be the assumption of hydrostatic balance in the dynamical equations. The technical details are described fairly fully in a set of appendices, and some typical results are presented in the later sections through applications of the tool to CMIP5 model runs under three different scenarios, representing pre-industrial conditions, historic anthropogenic forcing and a future "business as usual" projection. This serves well to illustrate the potential insights provided by the diagnostics on offer.

The paper overall is well structured and reasonably well written, although the English in a few places is a little "clunky" with occasional words missing or with awkward phrasing, which may tend to distract some readers. My main criticism (speaking as an "outsider" - not directly involved in diagnostic studies of state of the art Earth climate models) would be of the description in Section 2 of the data requirements for use of the tool. This is quite short and seems to presume quite a lot of knowledge on the part of the reader (unless they are prepared to spend a lot of time and effort consulting the documentation for the ESMValTool software, referenced as Eyring et al. (2016b)). Table 1 lists the variables required to be presented to the tool for the various calculations, but simply names the variables according to the CMOR convention without explanation. This is not helpful if you don't know that convention, and it is not explained anywhere in the paper except in external references. Why not provide a key for these variables or an explanation in another column of Table 1? Given the general applicability of much of the tool to climates that are not restricted to Earth (as acknowledged in the paper itself), readers would appreciate a bit more help and guidance for novice users. Is there a graphical interface, for example, or is it run via a shell script? Some more guidance here and perhaps an example script would be helpful, either in the text or another appendix.

*We thank the reviewer for the constructive criticism. We addressed the issue in two steps. We rearranged Table 1, in order to include a short description of each variable. In addition to that, we substantially rephrased the last part of Section 2, shortly explaining the algorithm. We did not dig into the details of the procedure, because the documentation of the code contains all the necessary information, either one uses it through the ESMValTool v.2, or prefers downloading the stand-alone version of the diagnostic tool (which is now available in a GitHub repository: https://github.com/ValerioLembo/TheDiaTo_v1.0.git)*

In other respects, this looks to be a potentially useful facility for anyone interested in model inter comparisons and/or complex climate diagnostics, and could be even more attractive if made a little more user-friendly. I outline below a few more minor points and suggestions.

- Figure 5 and associated text: The reservoir terms ASE, ATE, KSE and KTE are not explained or defined very obviously. They presumably refer to stationary and transient components of AE and KE? It would be helpful to state this in the caption and perhaps mention in the text?

*Thank you for the comment. We have modified the caption to Figure 5 and we added a short period at Section A2: "The contribution of eddies to APE (EPE) and KE (EKE) consists of two terms, the first one is the term accounting for stationary eddies (ASE and KSE in the diagrams of Figure 5, respectively). The second term accounts for the transient eddies (ATE and KTE in the diagrams of Figure 5, respectively). A discussion on the derivation of these terms is found in Ulbrich and Speth, 1991."*

- P.17 line 23 as example of awkward phrasing: should probably read ". . .as a possible reason for the well known cold pole bias. . .." But there are quite a few others – too numerous to list in detail. Perhaps recruit someone whose first language is English to proofread - or the authors should proofread more thoroughly before submission!

*We thank the reviewer for the constructive criticism. We have undergone a comprehensive proofread of the manuscript, together with a native speaker. We hope that this effort has substantially improved the quality of the language.*

- Presumably this tool could be used on reanalysis datasets too? Perhaps comment further?

*Thank you for the suggestion. We have added a sentence in the first paragraph of Section 2, explicitly mentioning that the software is also suitable for the ingestion of Reanalysis datasets: "Therefore, the tool is suitable for the evaluation of any kind of gridded datasets, provided that they contain the necessary variables on a regular grid, including blends of observations and Reanalyses. In our description of the software feature, we focus on model evaluation and multi-model intercomparison."*

---

## Author Response (AR1)

Dear editor,

referring to the manuscript :

"TheDiaTo (v1.0) – A new diagnostic tool for water, energy and entropy budgets in climate models" by Valerio Lembo, Frank Lunkeit and Valerio Lucarini (ID: gmd-2019-37)

please find attached the revised draft, with all tracked changes to the original document.

In the following, the comments raised by the two reviewers are reported together with the author's replies and, if applicable, the related changes in the text. Reviewers' comments are reported in plain text. Author's replies and modified text (with quotation marks) in italic.

In addition to that, we have modified Figures 1, 4-a, 6 in order to remove the blank vertical line on the GMT longitude. Labels in Figures 2 and 8 were modified as well, because the definition of enthalpy is more suitable to the computations that are performed here.

Yours Sincerely,

Valerio Lembo
* * *
**Referee RC1**

- Introduction: For energy and water budget studies using observations, please add L'Ecuyer et al, 2015 and Rodell et al, 2015. For studies evaluating the energy and water budgets in climate models, please add Demory et al, 2014; Terai et al, 2017; Vanniere et al, 2018 (e.g. in P2 L29, P3 L22, P4 paragraph 1.2). These studies highlighted the need for more robust evaluation of these budgets in climate models.

*Thank you for the suggestions. We have added these references in a new sentence in Sect. 1.2: "Water mass budget has been assessed in observations (L'Ecuyer et al., 2015; Rodell et al., 2015), as well as in climate models, focusing on the hydrological cycle alone (Terai et al., 2018) or evaluating it in conjunction with the energy budgets (Demory et al., 2014; Vannière et al., 2019)."*

*(l. 15-17, p. 4): "Water mass budget has been assessed in observations (L'Ecuyer et al., 2015; Rodell et al., 2015), as well as in climate models, focusing on the hydrological cycle alone (Terai et al., 2018) or evaluating it in conjunction with the energy budgets (Demory et al., 2014; Vannière et al., 2019)."*

- section 2 data and software requirements: are calculations performed on the native grid of the models or on a common grid?

*The calculations are performed on the native grid of the model, as provided in the CMIP repository. We specified this in Sect. 2 by adding the following sentence: "For model inter-comparisons (Section 5), computations are performed on the native grid of the model."*

*(l. 13-14, p. 6) " For model inter-comparisons (Section 5), computations are performed on the native grid of the model".*

- section 3: can the tool consider observational data, where available, to validate the models?

*The tool can be used to compare models and observational data, as long as the data are global-scale and on a regular grid. The ESMValTool v.2 framework allows for straightforward comparison with Reanalyses (as it will be discussed in the ESMValTool v.2 report that is soon to be submitted on this same journal). Since referee #1 also suggested to mention that, we have added a comment on the range of applicability to observational-based gridded datasets in Sect. 2 as follows: "Therefore, the tool is suitable for the evaluation of any kind of gridded datasets, provided that they contain the necessary variables on a regular grid, including blends of observations and Reanalyses. In our description of the software features, we focus on model evaluation and multi-model intercomparison."*

*(l. 5-8, p.g 6) "Therefore, the tool is suitable for the evaluation of any kind of gridded datasets, provided that they contain the necessary variables on a regular grid, including blends of observations and Reanalyses. In our description of the software features, we focus on model evaluation and multi-model intercomparison."*

- section 3: can the tool also work with seasonal means?

*Technically, no constraint is given on the usage of a subset of months. Although this would cause in principle no problem on the computations of the budgets and the LEC strength, the material entropy production would be inconsistent, because there is no straightforward argument justifying that the indirect method would provide a reasonable estimate of this quantity (as mentioned when discussing Eq. 11 in Sect. 3.4). A thorough explanation of the assumptions is provided in Lucarini et al. 2011. Furthermore, the implied transports rely on the fact that the differential heating is balanced by the oceans and the atmosphere, but this is not certainly accurate at the smaller than annual timescale (cfr. for instance Rose and Ferreira 2013 on the validity of the Bjerknes compensation mechanism).*

- section 3.2 hydrological cycle: Assessing the water budget in climate models is particularly important to determine their conservation error and its evolution throughout the simulation. A model may have a conservation error but that remains stable over time, while another one may constantly lose/gain mass over time (as noted in Liepert and Previdi, 2012). In this study, this would be particularly important to understand what is happening in BNU. Is it possible to see the time evolution of the conservation error with TheDiaTo (Fig. 2 of Liepert and Previdi), and if not, would it be possible to add it? This would be beneficial for a thorough evaluation of the climate models.

*The diagnostic tool allows to obtain annual mean time series of energy and water mass budgets for the Norther Hemisphere, the Southern Hemisphere and the global mean. For sake of brevity, we decided not to include them. An example of what the time series for the BNU-ESM model would look like is shown in Figures 1-3. It can be noticed that the conservation error is quite massive, compared to the other models, but relatively constant.*

**Figure 1:** *annual mean time series of the energy budget for BNU-ESM pre-industrial run. Red denotes Southern Hemisphere, blue denotes Northern Hemisphere, black the global mean.*

[Figure]

Figure 2: same as Figure 1, for the last 20 years of the historical experiment.

[Figure]

Figure 3: same as Figure 1, for the last 20 years of the RCP8.5 experiment.

- P15 L21: is there a reason for picking this period: 2441-2460?

*The reason for that is that there is no other section of the pre-industrial CanESM2 run for which 20-years of all the needed variables is available on the repository. We explained this by adding the following sentence: "The choice of the sub-period is motivated by the fact that it is the only part of the run for which a 20-years subsequent dataset of the needed variables is available in the repository."*

*(l. 14-15, p. 16) "The choice of the sub-period is motivated by the fact that it is the only part of the run for which a 20-years subsequent dataset of the needed variables is available in the repository".*

- P18 L28 & L32: 'bias' generally refers to errors relative to observations. Here, it may be more

robust to refer to 'imbalance'.

*Thank you. We replaced "bias" with the more appropriate term "imbalance".*

*(l. 11-12, p. 19) "As shown in the first two columns of Table 2, there is a large imbalance in the TOA energy budget under unforced piC conditions." (and elsewhere in sec. 5.1, l. 11-29)*

- Tables 8 and 9: specify that rows are for PiC, hist and rcp85.

*We have changed the table captions as follows: "For each model, the first row denotes the estimates from piC, the second row from hist, the third row from rcp8.5".*

*(Table 8 & Table 9 captions): "For each model, the first row denotes the estimates from piC, the second row from hist, the third row from rcp8.5. Values are in mW/m2K-1."*

- P1 L5: replace 'Top-of-Atmosphere (TOA)' by TOA, as defined just earlier

*Done. Thank you.*

*(l. 4-5, p. 1) "The routine takes as inputs energy fluxes at surface and at the Top-of-Atmosphere (TOA), which allows for the computation of energy 5  budgets at the TOA, the surface, and in the atmosphere as a residual."*

- P2 L16: remove 'of things'

*Done. Thank you.*

*(l. 16 p. 2): " Difficulties can only increase when looking at the practical side:"*

- P2 L21: be more specific with 'the required standards mentioned above' so we don't have to jump back to recall these

*We have replaced "prepared according to the required standards mentioned above" with "containing process-based and end-user-relevant diagnostics".*

*(l. 20-22, p. 2): "What we propose in this paper is a novel software, containing process-based and end-user-relevant diagnostics, capable of providing an integrated perspective on the problem of models validation and intercomparison."*

- P2 L23: same as above for 'enunciated above ideas'

*We have added "reaching a steady state as a result of an interplay of multi-scale processes".*

*(l. 22-24, p. 2): "The goal here is to examine models through the lens of their dynamics and thermodynamics, in the view of enunciated above ideas about complex non-equilibrium systems reaching a steady state as a result of an interplay of multi-scale processes."*

- P5 L5: same as above for 'mentioned aspects'

*We have added "(i.e. the energy and water mass budgets and meridional transports, the LEC strength and the material entropy production)".*

*(9-11, p. 3): "We present here TheDiaTo v1.0, a new software for diagnosing the mentioned aspects of thermodynamics of the climate systems (i.e. the energy and water mass budgets and meridional transports, the LEC strength and the material entropy production) in a broad range of global-scale gridded datasets of the atmosphere."*

- P23 L17: 'effected' -> 'affected'

*What we mean here is that baroclinic eddies "effect" baroclinic conversions. They can be both "synoptic-scale" and "planetary-scale" eddies, as shown in a recent paper of us (Lembo et al. 2019, 46, Geophys. Res. Lett.). We have added this reference, since it was not yet accepted by the time the manuscript was submitted.*

*(l. 29-31 p. 23) "The two quantities are related through the meridional enthalpy transports (cfr. Boschi et al. (2013); Lembo et al. (2019b)), which in mid-latitudes are mainly effected by baroclinic eddies."*

**Referee RC2**

My main criticism (speaking as an "outsider" - not directly involved in diagnostic studies of state of the art Earth climate models) would be of the description in Section 2 of the data requirements for use of the tool. This is quite short and seems to presume quite a lot of knowledge on the part of the reader (unless they are prepared to spend a lot of time and effort consulting the documentation for the ESMValTool software, referenced as Eyring et al. (2016b)). Table 1 lists the variables required to be presented to the tool for the various calculations, but simply names the variables according to the CMOR convention without explanation. This is not helpful if you don't know that convention, and it is not explained anywhere in the paper except in external references. Why not provide a key for these variables or an explanation in another column of Table 1? Given the general applicability of much of the tool to climates that are not restricted to Earth (as acknowledged in the paper itself), readers would appreciate a bit more help and guidance for novice users. Is there a graphical interface, for example, or is it run via a shell script? Some more guidance here and perhaps an example script would be helpful, either in the text or another appendix.

*We thank the reviewer for the constructive criticism. We addressed the issue in two steps. We rearranged Table 1, in order to include a short description of each variable. In addition to that, we substantially rephrased the last part of Section 2, shortly explaining the algorithm. We did not dig into the details of the procedure, because the documentation of the code contains all the necessary information, either one uses it through the ESMValTool v.2, or prefers downloading the stand-alone version of the diagnostic tool (which is now available in a GitHub repository: https://github.com/ValerioLembo/TheDiaTo_v1.0.git)*

*(l. 3-13, p. 7): "The ESMValTool architecture is implemented as a Python package and the latest version is available at www.doi.org/10.17874/ac8548f0315, where the dependencies and installation requirements are also described. If using TheDiaTo v.1 as part of the ESMValTool architecture, the user is asked to specify the path to input data, work and plot directories, as well as*

*some details of the local machine. A dedicated namelist (named as "recipe") includes information on the diagnostics, such as a brief description, the reference literature, the developers. The user is asked to specify here the name of the dataset to be analysed, that must be recognised by the ESMValTool preprocessor. The diagnostic tool is thus run calling the ESMValTool interface with the associated recipe.*
*A stand-alone version of the software is maintained as well, utilizing Python bindings for Climate Data Operators (CDO) (CDO, 2015). It consists of a Python script, reading a namelist with user settings (such as directory paths, model names and options for the usage of the modules). The stand-alone version is publicly available in a GitHub repository (https://github.com/ValerioLembo/TheDiaTo_v1.0), where detailed instructions on the usage of the software are given." and added column in Table 1.*

- Figure 5 and associated text: The reservoir terms ASE, ATE, KSE and KTE are not explained or defined very obviously. They presumably refer to stationary and transient components of AE and KE? It would be helpful to state this in the caption and perhaps mention in the text?

*Thank you for the comment. We have modified the caption to Figure 5 and we added a short period at Section A2: "The contribution of eddies to APE (EPE) and KE (EKE) consists of two terms, the first one is the term accounting for stationary eddies (ASE and KSE in the diagrams of Figure 5, respectively). The second term accounts for the transient eddies (ATE and KTE in the diagrams of Figure 5, respectively). A discussion on the derivation of these terms is found in Ulbrich and Speth, 1991."*

*(l. 6-9, p. 27): "The contribution of eddies to APE (EPE ) and KE (EKE ) consists of two terms, the first one is the term accounting for stationary eddies (ASE and KSE in the diagrams of Figure 5, respectively). The second term accounts for the transient eddies (ATE and KTE in the diagrams of Figure 5, respectively). A discussion on the derivation of these terms is found in Ulbrich and Speth (1991)."*

- P.17 line 23 as example of awkward phrasing: should probably read ". . .as a possible reason for the well known cold pole bias. . .." But there are quite a few others – too numerous to list in detail. Perhaps recruit someone whose first language is English to proofread - or the authors should proofread more thoroughly before submission!

*We thank the reviewer for the constructive criticism. We have undergone a comprehensive proofread of the manuscript, together with a native speaker. We hope that this effort has substantially improved the quality of the language.*

*(l. 4-6, p. 18): "This is consistent with previous findings (cfr. Boer and Lambert (2008); Marques et al. (2011)) and is possibly due to the well-known cold Pole bias and the consequently excessive speed of the jet stream." Generally, as evidenced in the document with tracked changes, we performed a careful rewording of the manuscript.*

- Presumably this tool could be used on reanalysis datasets too? Perhaps comment further?

*Thank you for the suggestion. We have added a sentence in the first paragraph of Section 2, explicitly mentioning that the software is also suitable for the ingestion of Reanalysis datasets: "
[revised manuscript text omitted]